# Adversarial Robustness of Graph Transformers

## Abstract

Existing studies have shown that Message-Passing Graph Neural Networks (MPNNs) are highly susceptible to adversarial attacks. In contrast, despite the increasing importance of Graph Transformers (GTs), their robustness properties are unexplored. Thus, for the purpose of robustness evaluation, we design the first adaptive attacks for GTs. We provide general design principles for strong gradient-based attacks on GTs w.r.t. structure perturbations and instantiate our attack framework for five representative and popular GT architectures. Specifically, we study GTs with specialized attention mechanisms and Positional Encodings (PEs) based on random walks, pair-wise shortest paths, and the Laplacian spectrum. We evaluate our attacks on multiple tasks and threat models, including structure perturbations on node and graph classification and node injection for graph classification. Our results reveal that GTs can be *catastrophically fragile* in many cases. Consequently, we show how to leverage our adaptive attacks for adversarial training, substantially improving robustness.

## 1 Introduction

Graphs are versatile data structures that have applications in a wide range of different domains, and Graph Neural Networks (GNNs) have become the tool of choice for many learning tasks on graphs. Given the increasing popularity of GNNs, multiple studies in the past years have developed adversarial attacks for GNNs and analyzed their robustness (Zügner et al., 2018; Zügner & Günnemann, 2019; Zügner & Günnemann, 2020). These studies mostly focus on Message-Passing GNNs (MPNNs), such as the Graph Convolutional Network (GCN) (Kipf & Welling, 2017) and show that GNNs are vulnerable to even slight graph structure perturbations (Zügner et al., 2018).

Recently, Graph Transformers (GTs) have gained popularity (Müller et al., 2024), addressing inherent limitations such as *over-smoothing*, *over-squashing*, and limited receptive fields (Müller et al., 2024). Yet, the adversarial robustness of GTs is entirely unexplored. The unknown stability of GTs poses a substantial risk in practical applications, where robustness is crucial. However, evaluating the robustness of GTs is non-trivial. GTs' employ modified attention mechanisms and Positional Encodings (PEs) that often include parts that are not differentiable w.r.t. the graph structure. This renders the application of gradient-based adaptive attacks difficult, even though the gradient is key for crafting adversarial attacks efficiently. Adaptive attacks can use the model-specific gradient to adjust to all architecture details and are vital for realistic robustness estimates (Athalye et al., 2018; Carlini & Wagner, 2017; Tramèr et al., 2020), also in the GNN domain (Mujkanovic et al., 2022).

We provide the first analysis of the robustness of GTs. To obtain precise robustness estimates, we propose continuous relaxations and perturbation approximations for the most widely used GT components including **(a) Shortest Path**, **(b) Random Walk**, and **(c) Spectral** PEs. These enable us to apply adaptive gradient-based attacks to five popular and representative GT architectures: **1)** Graph Inductive bias Transformer (**GRIT**) (Ma et al., 2023), **2) Graphormer** (Ying et al., 2021), **3)** Spectral Attention Network (**SAN**) (Kreuzer et al., 2021), **4)** General, Powerful, Scalable (**GPS**) GT (Rampášek et al., 2022), and **5) Polynormer** (Deng et al., 2024). Moreover, we provide guiding principles to develop relaxations for other discrete or non-differentiable components in GTs.

Our study reveals that GTs can be catastrophically fragile if evaluated with our adaptive attacks (Fig. 1). For example, on the proposed node injection attacks (NIAs) for fake news detection (Fig. 1c & 1d), perturbing $2.5\%$ of edges suffices to essentially halve the accuracy. Consequently, we use

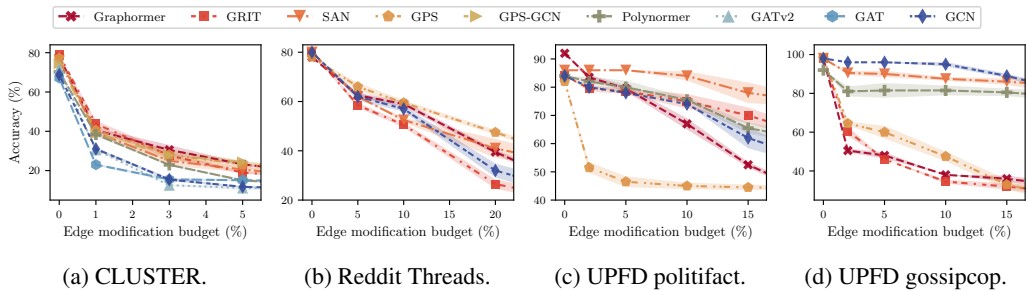

(a) CLUSTER.  (b) Reddit Threads.  (c) UPFD politifact.  (d) UPFD gossipcop.

Figure 1: The adversarial classification accuracy for different GNNs with varying (evasion) attack budgets on four different datasets: CLUSTER - inductive node classification (global structure attack), Reddit Threads - graph classification (structure attack), UPFD politifact and gossipcop - graph classification (node injection attack) . The strongest attack for each budget is shown.

our adaptive attacks to devise an effective adversarial training strategy that largely alleviates the hypersensitivity of some GT architectures.

Our main contributions are: **(1)** We formulate adaptive gradient-based structure attacks, along with the guiding principles, for five Graph Transformer (GT) architectures. **(2)** We are the first to study the adversarial robustness of graph transformers and show that they suffer from similar vulnerabilities to adversarial attacks as message-passing GNNs. **(3)** We show that adversarial training yields an effective defense counteracting GT's vulnerabilities.

## 2 BACKGROUND

Let $\mathcal{G} = (\mathcal{V}, \mathcal{E})$ be an undirected attributed graph with $n$ nodes $\mathcal{V} = \{v_1, ..., v_n\}$ and $m$ edges. Let $\boldsymbol{x}_i \in \mathbb{R}^d$ be the feature vector of node $v_i$. Then the graph can be defined as $\mathcal{G} = (\boldsymbol{A}, \boldsymbol{X})$ with its symmetric binary adjacency matrix $\boldsymbol{A} \in \{0, 1\}^{n \times n}$ and node feature matrix $\boldsymbol{X} \in \mathbb{R}^{n \times d}$. The diagonal degree matrix $\boldsymbol{D}$ with entries $\boldsymbol{D}_{ii} = \deg(v_i) = \sum_{j=1}^{n} \boldsymbol{A}_{ij}$ and the normalized symmetric graph Laplacian matrix $\boldsymbol{L}_{sym} = \boldsymbol{I} - \boldsymbol{D}^{-1/2} \boldsymbol{A} \boldsymbol{D}^{-1/2}$ can both be derived from $\boldsymbol{A}$. The GNNs considered in this work are functions $f_\theta(\boldsymbol{A}, \boldsymbol{X})$ with model parameters $\theta$. We denote the updated hidden node representations after each GNN layer $l$ as $\boldsymbol{H}^{(l)}$ with initialization $\boldsymbol{H}^{(0)} = \boldsymbol{X}$. For node-level tasks, the output node representations are directly utilized for predictions, while for graph-level tasks a graph-pooling operation aggregates the nodes embeddings into a graph embedding before prediction.

### 2.1 STRUCTURE ATTACKS

In this work we focus on *untargeted* white-box *evasion* attacks, i.e., an attacker with full knowledge of model and data attempts to change the trained model's prediction to any incorrect class at test time by slightly perturbing the input graph structure. For node-level tasks we focus on *global* attacks that minimize the overall performance metric across all nodes. The attack objective is described by the following optimization problem:

$$\max_{\tilde{\boldsymbol{A}} \text{ s.t. } ||\tilde{\boldsymbol{A}} - \boldsymbol{A}||_0 < \Delta} \mathcal{L}_{atk}(f_\theta(\tilde{\boldsymbol{A}}, \boldsymbol{X})) \tag{1}$$

where $f_\theta$ is the GNN model with fixed parameters $\theta$, $\tilde{\boldsymbol{A}} \in \{0, 1\}^{n \times n}$ is the discrete perturbed adjacency matrix in relation to $\boldsymbol{A}$ with the number of edge flips bounded by the budget $\Delta$, and $\mathcal{L}_{atk}$ is a suitable attack loss function. For node classification, we use the *tanh-margin* attack loss proposed in (Geisler et al., 2021). For graph classification, we simply optimize the unnormalized class logits: $\mathcal{L}_{atk} = -l_y + \sum_{c \neq y} l_c$. It is convenient to model the perturbation as a function of the binary matrix indicating the edge flips $\boldsymbol{B} \in \{0, 1\}^{n \times n}$:

$$\tilde{\boldsymbol{A}} = \boldsymbol{A} + \delta \boldsymbol{A}, \qquad \delta \boldsymbol{A} = (\boldsymbol{1}_n \boldsymbol{1}_n^\mathsf{T} - 2\boldsymbol{A}) \odot \boldsymbol{B} \tag{2}$$

with element-wise product $\odot$. The combinatorial problem can be optimized more efficiently with a continuous relaxation $\boldsymbol{B} \in [0, 1]^{n \times n}$. In this setting, the entry $\boldsymbol{B}_{ij}$ represents the probability that the

edge $(v_i, v_j)$ is flipped. Discrete perturbations can then be sampled from the continuous solution. The budget constraint becomes $\mathbb{E}[\mathsf{Bernoulli}(\boldsymbol{B})] = \sum \boldsymbol{B}_{ij} \leq \Delta$, which can be dealt with by using projected gradient descent (Xu et al., 2019). For large graphs, optimizing over all entries in $\boldsymbol{B}$ at once becomes infeasible. Projected Randomized Block Coordinate Descent (PRBCD) solves this with a strategy that optimizes over sampled random blocks of limited size (Geisler et al., 2021).

## 2.2 GRAPH TRANSFORMERS

Graph transformers (GTs) apply the popular transformer architecture for sequences (Vaswani et al., 2017) to arbitrary graphs. A general GT architecture is depicted in Fig. 2. In this work, we focus on GTs that apply global self-attention, where each node can attend to all other nodes. A 'vanilla' structure-unaware self-attention head is defined as:

$$\mathsf{Attn}(\boldsymbol{H}) = \mathsf{softmax}\left(\frac{(\boldsymbol{H}\boldsymbol{W}_q)(\boldsymbol{H}\boldsymbol{W}_k)^{\mathsf{T}}}{\sqrt{d}}\right)(\boldsymbol{H}\boldsymbol{W}_v) \tag{3}$$

where $\boldsymbol{W}_q, \boldsymbol{W}_k, \boldsymbol{W}_v \in \mathbb{R}^{d \times d}$ are the weights for the *query*, *key*, and *value* projections. The individual attention scores can thus be defined as:

$$\alpha_{ij} = \mathsf{softmax}(w_{ij}) = \frac{e^{w_{ij}}}{\sum_k e^{w_{ik}}}, \quad \text{with} \quad w_{ij} = \frac{\boldsymbol{W}_q \boldsymbol{h}_i \cdot \boldsymbol{W}_k \boldsymbol{h}_j}{\sqrt{d}} \tag{4}$$

Since this update is independent of the graph structure, many GTs apply a modified attention mechanism that also depends on the adjacency matrix. However, the most common way to add structure information is by adding Positional Encodings (PEs) to the node features:

$$\boldsymbol{H}^{(0)} = \boldsymbol{X} + \psi(\boldsymbol{A}) \tag{5}$$

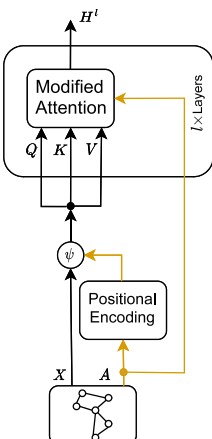

Figure 2: A generic graph transformer architecture.

We categorize the PEs roughly in three main categories: (1) random walk encodings, (2) distance encodings, and (3) spectral encodings. Next we describe the PEs and attention modifications of the five GT models that we attack. More details about the architecture are available in § C.

**Graphormer** (Ying et al., 2021). For the PEs, a degree embedding vector $\boldsymbol{z}_i \in \mathbb{R}^d$ is learned for each discrete node degree value. Each node feature gets added to a PE vector of its corresponding degree. Similarly, a scalar $b_i \in \mathbb{R}$ is learned for each discrete Shortest Path Distance (SPD). These values are added to the pair-wise raw attention scores according to the pairs' SPDs. This results in a re-weighting of the attention weights after applying the softmax function.

**GRIT** (Ma et al., 2023). For the PEs, the random walk probability matrices for walks of lengths 0 to $k-1$ are concatenated to form a 3D tensor. By slicing this tensor, an embedding vector $\boldsymbol{P}_{i,j,:} \in \mathbb{R}^k$ is obtained for each of the $n^2$ node-pairs $(v_i, v_j)$. The diagonal entries are projected by a linear layer and added to the node features as PEs. Additionally, all $n^2$ embedding vectors are used as node-pair features. The node representations $\boldsymbol{h}_i$ and node-pair representations $\boldsymbol{h}_{i,j}$ are both updated in each transformer layer by a modified attention mechanism, which includes scaling by the node degrees.

**SAN** (Kreuzer et al., 2021). The PEs are based on the eigen-decomposition of the Laplacian matrix. Specifically, for each node $v_i$ the PE is initialized as a sequence of $k$ vectors resulting from the concatenation of the $k$ smallest eigenvalues with the $i$-th entries of their corresponding eigenvectors. The sequence is further processed by a transformer encoder and then pooled in the sequence dimension. The resulting PE vectors are concatenated to the node features. Additionally, the main graph transformer attention mechanism is modified to have two separate key and query projection weights for connected and unconnected node-pairs. The attention scores to the connected nodes and to the unconnected nodes are computed independently, each with a softmax. A hyperparameter $\gamma$ controls how the two scores are relatively scaled, varying the bias towards sparse or full attention.

**GPS** (Rampášek et al., 2022). Each GPS layer combines local message passing of a MPNN with a global attention update. We consider the configuration with DeepSet spectral PEs (similar to SAN), a GatedGCN (Bresson & Laurent, 2018) MPNN, and standard transformer global attention. Alternatively, if a GCN (Kipf & Welling, 2017) is used as the MPNN we denote the model as GPS-GCN.

**Polynormer** (Deng et al., 2024). This model first applies local message-passing layers followed by linear global attention layers. The local layer is based on GAT (Veličković et al., 2018) and all updates are of second polynomial order. No PEs are used. Instead, the global attention relies on the previous message-passing steps to aggregate enough information about the graph structure.

## 3    ATTACKING GRAPH TRANSFORMERS

The main obstacles for gradient-based structure attacks on GTs $f_\theta$ are PEs and attention mechanisms that are designed for a discrete graph structure. As a result, the model output is often a discontinuous function of the continuous input adjacency matrix $\tilde{A} \in [0,1]^{n \times n}$, rendering continuous optimization ineffective. Thus, to obtain useful gradients we need to relax the structure aware components such as PEs. For designing the continuous relaxations $\tilde{f}_\theta$, we identify three main principles:

**Principle I: Relaxed and target models coincide for discrete inputs.** The prediction should equal $\tilde{f}_\theta(A) = f_\theta(A)$ for any discrete adjacency $A \in \{0,1\}^{n \times n}$.

**Principle II: $\tilde{f}_\theta$ can interpolate "smoothly" between any different discrete graphs.** In other words, $\tilde{f}_\theta(\tilde{A})$ should be continuous w.r.t. $\tilde{A}$ but does not need to be continuously differentiable.

**Principle III: The relaxed model $\tilde{f}_\theta$ must be efficient.** It is a critical property that the relaxation does not excessively increase memory and runtime complexity.

While *Principle II* might appear surprising at first glance, we argue that we do not need to enforce stronger standards on $\tilde{f}_\theta$ than perhaps the most widely used activation function ReLU. We expect these principles to be sensible defaults for the design of adaptive attacks on future GTs.

Some models, such as **GCN** and **GRIT**, are already continuous and do not require relaxations. Models with local attention, such as **GAT**, use the adjacency matrix to determine a binary attention mask from the local neighborhoods $\mathcal{N}(v_i) = \mathcal{N}_i = \{v_j | A_{ij} > 0\})$, while the attention weights themselves are computed using only the node features:

$$\alpha_{ij} = \mathsf{softmax}_{\mathcal{N}_i}(w_{ij}) = \frac{e^{w_{ij}}}{\sum_{v_k \in \mathcal{N}_i} e^{w_{ik}}}, \qquad w_{ij} = f_{\theta,attn}(\boldsymbol{h}_i, \boldsymbol{h}_j) \tag{6}$$

Note that the local neighborhood is discontinuous w.r.t. continuous changes in the adjacency matrix. As a continuous relaxation, we can instead add a bias to the attention weights: $\tilde{w}_{ij} = w_{ij} + g(\tilde{A}_{ij})$. Because of the exponentials in the softmax operation, to re-obatin the binary attention mask in the discrete case, we require $g(0) = -\infty$ and $g(1) = 0$. A simple valid choice is the natural logarithm. The attention scores can then be written as:

$$\tilde{\alpha}_{ij} = \mathsf{softmax}(w_{ij} + \log(\tilde{A}_{ij})) = \frac{e^{w_{ij} + \log(\tilde{A}_{ij})}}{\sum_k e^{w_{ij} + \log(\tilde{A}_{ij})}} = \tilde{A}_{ij} \cdot \frac{e^{w_{ij}}}{\sum_k \tilde{A}_{ik} \cdot e^{w_{ik}}} \tag{7}$$

As a results, the binary attention mask is emulated in the discrete case and in the continuous case the attention scores to nodes that are only loosely connected are decreased. Using this relaxation (and a similar one for GatedGCN), we define continuous **GPS** and **Polynormer** relaxations. Note that Eq. 7 is neither needed nor valid for global attention which can include contributions from all nodes in the graph regardless of connectivity.

**Graphormer**. The degree PEs $\boldsymbol{z}_{\mathsf{deg}(v_i)}$ and SPD biases $b_{\mathsf{spd}(v_i,v_j)}$ are indexed by the discrete values of the node degrees (# of neighbors) and shortest path distances (# of hops). To enable the use of continuous degrees, we define a linear interpolation between the PE embeddings of the two closest integer degree values:

$$\tilde{\boldsymbol{z}}_{\mathsf{deg}(v_i)} = \eta \cdot \boldsymbol{z}_{\mathsf{d}_l + 1} + (1 - \eta) \cdot \boldsymbol{z}_{\mathsf{d}_l}, \quad \text{with} \quad \mathsf{d}_l = \lfloor \mathsf{deg}(v_i) \rfloor, \quad \eta = \mathsf{deg}(v_i) - \mathsf{d}_l \tag{8}$$

Increasing the edge probabilities to a node also increases the expected discrete degree. However, the edge probabilities are more challenging to interpret for the SPDs. When a very small edge probability lies on a (simple) shortest path, the path is less likely to exist in the discrete sampled adjacency matrix. Therefore, low edge probabilities should only marginally affect the original SPDs. To model this relationship, we use the reciprocal of the adjacency matrix $\boldsymbol{R}_{ij} = {}^{1\!}/{}_{\tilde{A}_{ij}}$ to find

continuous proxy shortest path distances $\mathsf{rspd}_{ij} = \mathsf{spd}(v_i, v_j | \boldsymbol{R})$. We interpolate between the closest discrete values again and obtain:

$$\tilde{b}_{\mathsf{spd}(v_i, v_j)} = \eta \cdot b_{\mathsf{s}_l+1} + (1 - \eta) \cdot b_{\mathsf{s}_l}, \quad \text{with} \quad \mathsf{s}_l = \lfloor \mathsf{rspd}_{ij} \rfloor, \quad \eta = \mathsf{rspd}_{ij} - \mathsf{s}_l \quad (9)$$

Note that for discrete edge probability values 0 and 1, the reciprocal edge weights become $-\infty$ and 1, respectively, yielding the original SPDs. Hence, we do not alter the clean predictions if $\delta \boldsymbol{A} = \boldsymbol{B} = \boldsymbol{0}$ (see *Principle I*).

**SAN**. Since SAN applies a separate attention mechanism based on the binary decision whether an edge is real or fake, it can be interpreted as applying local attention to the original graph and to the 'inverse' graph in parallel. Analogously to Eq. 7, we bias the 'real' and 'fake' edge attention score with $\tilde{w}_{\mathsf{real},ij} = w_{\mathsf{real},ij} + \log(\tilde{\boldsymbol{A}}_{ij})$ and $\tilde{w}_{\mathsf{fake},ij} = w_{\mathsf{fake},ij} + \log(1 - \tilde{\boldsymbol{A}}_{ij})$ respectively. Note that for a discrete real edge, i.e., $\tilde{\boldsymbol{A}}_{ij} = 1$, the logarithm terms become 0 and $-\infty$ respectively, such that it fully contributes to the 'real' attention mechanism while not affecting the 'fake' one. The same can be shown for discrete fake edges, thus the descrete output remains unchanged (see *Principle I*).

The Laplacian matrix itself is a continuous functions of the entries in the adjacency matrix. However, its eigen-decomposition used for the PEs poses some challenges for gradient computation, especially w.r.t. the eigenvectors. The problems arise because: (a) the choice of direction (sign) for eigenvectors is arbitrary, (b) the choice of an eigenvector-basis of the eigenspace of a repeated eigenvalue is arbitrary, thus the gradient is not well defined, (c) for eigenvalues that are close together, the corresponding eigenvector gradients are numerically unstable. To avoid direct gradient computation, we use results from matrix perturbation theory (Stewart & Sun, 1990; Bamieh, 2022) to approximate the perturbed eigen-decomposition as a simpler function of the input perturbation. We define the perturbation on the Laplacian as $\delta \boldsymbol{L}_{sym} = \tilde{\boldsymbol{L}}_{sym} - \boldsymbol{L}_{sym}$, where $\tilde{\boldsymbol{L}}_{sym}$ is the Laplacian of the perturbed continuous adjacency matrix $\tilde{\boldsymbol{A}}$. The first-order approximations from (Bamieh, 2022) are, for the eigenvalues and eigenvectors:

$$\tilde{\boldsymbol{\Lambda}} = \boldsymbol{\Lambda} + \delta \boldsymbol{\Lambda}, \quad \delta \boldsymbol{\Lambda} \approx \mathsf{diag}(\boldsymbol{U}^\top \delta \boldsymbol{L} \, \boldsymbol{U}) \quad (10)$$

$$\tilde{\boldsymbol{U}} = \boldsymbol{U} + \delta \boldsymbol{U}, \quad \delta \boldsymbol{U} \approx -\boldsymbol{U} \left( \boldsymbol{\Pi} \odot \left( \boldsymbol{U}^\top \delta \boldsymbol{L} \, \boldsymbol{U} \right) \right), \quad \text{where} \quad \boldsymbol{\Pi}_{ij} = \begin{cases} \frac{1}{\lambda_i - \lambda_j} & \text{if } \lambda_i \neq \lambda_j \\ 0 & \text{else} \end{cases} \quad (11)$$

However, when repeated eigenvalues are present in the unperturbed Laplacian, special care for the choice of the eigenvectors in $\boldsymbol{U}$ that span the eigenspaces of the repeated eigenvalues is required. This case is treated by Bamieh (2022) and we show the application in our specific case in § H.1. A completely different strategy consists of adding a bit of random noise to $\tilde{\boldsymbol{L}}_{sym}$ in hopes of breaking apart any repeated eigenvalues, such that is possible to directly backpropagate through the eigendecomposition Lin et al. (2022). We elaborate on this strategy and propose an alternative in § H.2.

## 4 NODE INJECTION ATTACK

We also consider the relevant case of inserting nodes into an existing graph structure. In contrast to the usual framing of Node Injection Attack (NIA), where the attacker also chooses the node features for the new *vicious* nodes (Wang et al., 2020), we connect existing nodes from other graphs of an inductive graph dataset. Therefore, the nodes' features are fixed but physically realizable even if, e.g., they represent embeddings of natural language. This alleviates us from a somewhat subjective definition of imperceptibility required to craft the node features in the existing NIA. Hence, our attack solely focuses on 'structure' perturbations and their influence on the PEs, which are of particular interest for attacking GTs.

We formulate our node injection attack as a structure attack on an augmented graph that includes both the original nodes and the set of potential injection nodes. This formulation enables the use of the same PRBCD attack optimization, where the edge flip budget constraint also serves as an upper bound for the number of nodes that can be injected. We provide details in D.

**Node probability for smooth node insertion**. The continuous optimization of structure attacks in § 2.1 assigns probabilities to edges-flips, while nodes are assumed to all be part of the graph. In contrast, during NIAs the nodes also have have certain probabilities of being included. To approximate these node probabilities from the edge weights in a general way, we propose a simple iterative

computation. For a continuous connected graph, let $\mathcal{N}(v_i)$ be the 1-hop neighborhood of node $v_i$. We can calculate the probability $p_i$ of $v_i$ being connected to the graph, by using the probability of being connected to its neighbors and the probabilities that these neighbor themselves are connected to the graph. We start with the assumption that all nodes are connected to the graph and update using the edge probabilities:

$$p_i^{(t+1)} = 1 - \prod_{v_j \in \mathcal{N}_i} (1 - \boldsymbol{A}_{ij} \cdot p_j^{(t)}), \quad \text{with} \quad p_i^{(0)} = 1 \tag{12}$$

An example is shown in § D.1. To ensure that the model output is continuous w.r.t. node injections, this node probability is used to compute a weighted sum or mean in the graph-pooling for graph level tasks. Additionally for GTs, we use the node probability to bias the global pairwise attention scores which result in a continuous weighting of the attention scores similar to Eq. 7:

$$\hat{w}_{ij} = w_{ij} + \log(p_j), \qquad \hat{\alpha}_{ij} = \mathsf{softmax}(\hat{w}_{ij}) = \frac{e^{\hat{w}_{ij}}}{\sum_k e^{\hat{w}_{ik}}} = \frac{p_j \cdot e^{w_{ij}}}{\sum_k p_k \cdot e^{w_{ik}}} \tag{13}$$

Note that while Eq. 7 is only valid for local attention, the potentially complete removal or addition of a node has global influence. Therefore, this node probability bias can be applied to any global attention mechanism.

## 5 EVALUATION

**Datasets**. We first evaluate our structure attacks on *CLUSTER* (Dwivedi et al., 2023). It contains SBM-generated graphs with 6 clusters, where each cluster has one labeled node. The average number of nodes is 117.2. The task is to predict which node belongs to which cluster, i.e., inductive node classification. For training, we use the standard PyG train/val/test split of 10000/1000/1000 graphs, respectively. Additionally, we evaluate on the graph classification dataset *Reddit Threads* (Rozemberczki et al., 2020). It contains 203 088 small graphs without any features and with an average of 23.9 nodes. The graphs represent users that are connected if they directly reply to each other in the thread. The task is binary classification of whether the thread is disscussion-based. We use stratified random train/val/test split of 75%/12.5%/12.5%.

We use the *UPFD* Twitter fake news detection datasets from (Dou et al., 2021) to evaluate our node injection attacks. There are 2 datasets: *politifact*, with political; and *gossipcop* with celebrity fake news. The average number of nodes is 131 and 58, respectively. The graphs consist of retweet trees, where the root node has features concerning both the news content and the user who posted the news. All other nodes' features are related to the users that retweeted the news. We add all dataset nodes except for the graph roots into the candidate set of injection nodes, since the roots are special. Moreover, we do not allow for perturbations of the original tree structure and if the discretely sampled injection perturbations do not have a tree structure, we take the maximum spanning tree (using the edge probabilities) to ensure all perturbations are valid retweet trees. The task is binary classification whether the graph contains fake news or not. We use the standard PyG train/val/test split of 20/10/70% of the 314 and 5464 graphs for politifact and gossipcop, respectively.

Due to the quadratic scaling in the number of nodes of the three chosen GTs, their application is limited to smaller graphs. This renders evaluation on larger graph datasets commonly used in robustness studies impractical. While GTs are most widely applied to molecule data, adversarial attacks are of little practical relevance in that domain. Thus, we omit molecule data from our evaluations.

**Attacks**. As explained in § 2.1, we apply *untargeted (global) evasion* attacks, i.e., we perturb the graph structure of the test input for a trained model with fixed weights. For model training we do a random hyperparameter search, choosing the model with the highest validation metric. This approach is consistent with common practice. The hyperparameters used for the attacks are reported in § G. We show results for 4 different attacks. *Adaptive PRBCD* uses our relaxations described in § 3 for a gradient-based PRBCD attack. *Random perturbation* is the simplest baseline, where a single random perturbation of the adjacency matrix is used. In contrast, *random attack* is a brute-force random search that tests many random perturbations and selects the best. To match the computational budget of the adaptive attacks, it gets the same number of model evaluations. Finally, the *GCN PRBCD transfer* attack transfers the perturbation computed from a PRBCD attack of a GCN model to the GT models. This is a strong baseline attack that follows the same principle

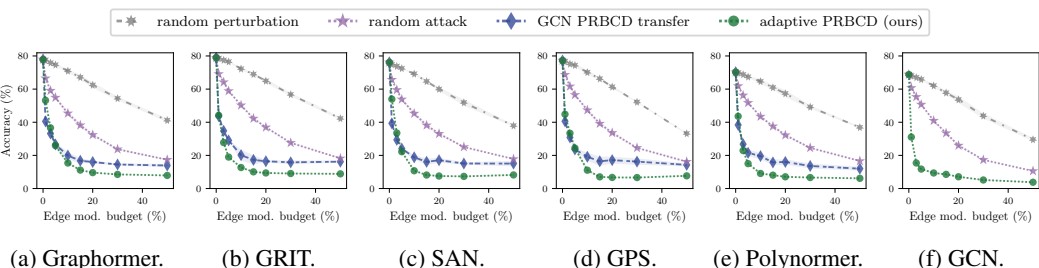

Figure 3: Global structure (evasion) attack results for CLUSTER (inductive node classification).

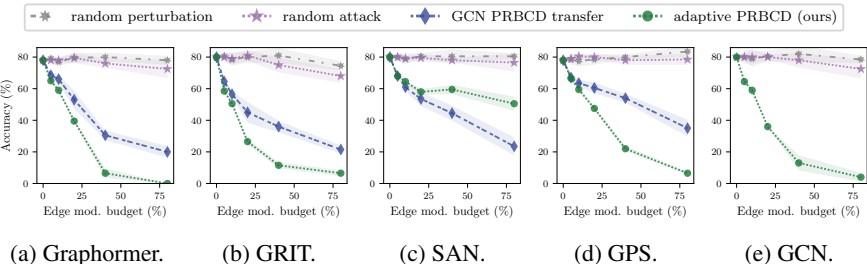

Figure 4: Structure (evasion) attack results for Reddit Threads (graph classification).

as many other established GNN attacks for different settings, such as Nettack (Zügner et al., 2018) and Mettack (Zügner & Günnemann, 2019): it is a gradient-based attack (PRBCD) on a simpler surrogate (GCN) that gets transferred to the victim model. Moreover, it is the main (global evasion) attack for non-GCN models proposed and used by (Geisler et al., 2021), where it has been show to be just as effective (or more) than other baselines.

For all datasets, we evaluate our attacks on the 50 first graphs in the test set and report average and standard deviation over 4 random seeds. For UPFD node injection, we use a small block size of 1000, which is necessary due to the $n^2$ scaling of GTs. For GCN it is possible to increase the block size, but we keep it the same for comparability. We optimize all our adaptive attacks for 125 steps and sample 20 discrete perturbations from the result, of which we take the strongest. For all other attack hyperparameters, we use default values that performed well in preliminary evaluations. For all results outside of the ablation tables, we use all of our continuous relaxations proposed in § 3.

## 6 ATTACK RESULTS

We present the first principled analysis on the robustness of GTs on five representative architecture types (Graphormer, GRIT, SAN, GPS, Polynormer). We define different goals for our evaluation: **(A)** efficacy of the proposed adaptive attacks, **(B)** providing an accurate assessment of GT robustness for relevant real-world tasks. To this end, we perform our evaluation on datasets with varying complexity. Towards **(A)** we explore the robustness of GTs on CLUSTER and Reddit Threads, which comprise simple, interpretable structures. This exploration helps us evaluate the effectiveness of the proposed relaxations, ideally leading our attacks to target semantically meaningful structures within the dataset. We address **(B)** through evaluations on UPFD. Here, we constrain our attack to remain within the predefined tree structure of the dataset. As a result, the attack represents impersonating an existing user who is retweeting the respective news article. This evaluation goes beyond previous robustness analyses of citation networks in GNNs (Zügner et al., 2018; Geisler et al., 2021), offering a more practical use case and semantically meaningful attacks.

**CLUSTER**. The node classification accuracy for the CLUSTER dataset for different attack budgets is shown in Fig. 3. The fragility of the data can already be seen by the *random perturbation* attack. Since only a single node in each cluster is labeled, attacking these labeled nodes requires little budget and leads to strong attacks. We manually inspected the adaptive attack perturbations and confirmed that most edge modifications are connected to the labeled nodes, which shows their efficacy. The GCN transfer attacks tend to work very well, strengthening our hypothesis that the straightforward nature of the task leads to the same type of semantically meaningful model-independent perturba-

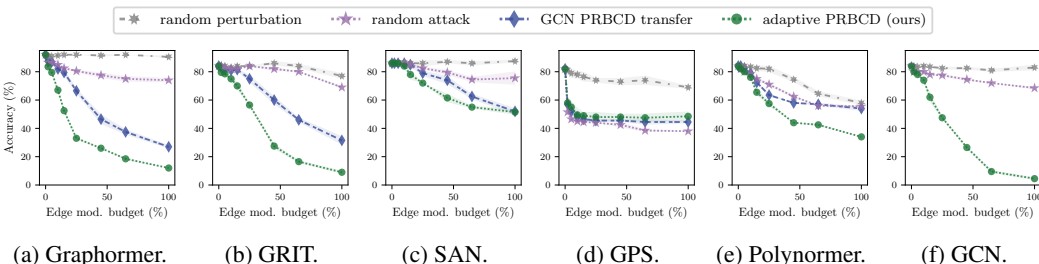

(a) Graphormer. (b) GRIT. (c) SAN. (d) GPS. (e) Polynormer. (f) GCN.

Figure 5: Node injection (evasion) attack results for UPFD politifact (graph classification).

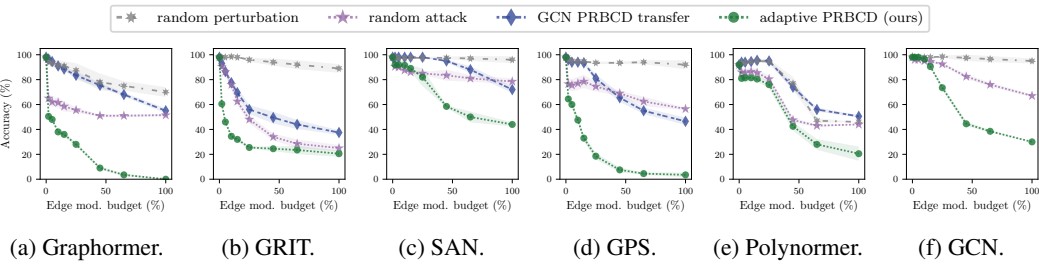

(a) Graphormer. (b) GRIT. (c) SAN. (d) GPS. (e) Polynormer. (f) GCN.

Figure 6: Node injection (evasion) attack results for UPFD gossipcop (graph classification).

tions: modifying edges to labeled nodes. This outcome positively indicates the effectiveness of our adaptive attacks **(A)**, as they consistently identify meaningful perturbations across all GTs. They may sometimes be weaker than the GCN transfer baseline (for the smallest budgets) simply due to a more difficult optimization function. To avoid the natural fragility in the data, we also evaluate a constrained attack that prohibits modifying edges to the labeled nodes, for which results are shown in § E.1. For CLUSTER, we additionally attack the GAT (Veličković et al., 2018) and GATv2 (Brody et al., 2022). The worst case perturbation results for all models are shown in Fig 1a, where we observe that GTs are consistently slightly more robust than MPNNs for small budgets (their clean accuracies are also higher though).

**Reddit Threads**. For this dataset we were unable to train a comparable Polynormer model. Fig. 1b shows a comparison of the models' robustess for small budgets. While there are differences in robustness, all models follow a similar trend. In Fig. 4, the individual attack accuracies for a wider range of budgets are shown for all models. While the GCN transfer attacks are also effective, our adaptive attacks are significantly stronger and the adversarial accuracy drops close to zero when up to 75% of the edges can be modified. SAN is the exception, for which in this particular case the adaptive attacks are comparatively weak. The low accuracy for large budgets is unsurprising, as there are no node features and the prediction relies soly on the graph structure. Interestingly, the random attack never seems to work well. This demonstrates that the gradient information provided by our relaxations is extremely helpful for finding good perturbations.

**UPFD**. The graph classification accuracies for different attack budgets are shown in Fig. 5 and Fig. 6 for the politifact and gossipcop dataset. In contrast to the results observed for the previous datasets, we note big robustness differences across models. In general, Graphormer seems to be the least robust and for which our adaptive attacks work best in comparison to the baselines. In most cases the adaptive attacks are the strongest, providing the best estimates of the models' robustness, highlighting the efficacy and importance of our gradient-based adaptive attacks. Moreover, these results reveal that GTs can showcase catastrophic vulnerabilities to adversarial modifications of the graph structure, even when these changes are constrained to meaningful perturbations. Fig. 1 provides a direct model comparison of the worst case perturbations for smaller budgets. It shows that the GCN model can exhibit considerably higher robustness than some GTs. The SAN model is the exception, as it is suprisingly robust for both datasets.

**Transferability**. We collected the adversarial examples generated for each of our adaptive GT attacks and applied them to the other models. In Fig. 7, we compare the strongest such transfer attack (*best transfer*) with the *GCN transfer* and *adaptive* attacks on UPFD gossipcop. The results show that our GT attack perturbations transfer better than from GCN. This may be because the

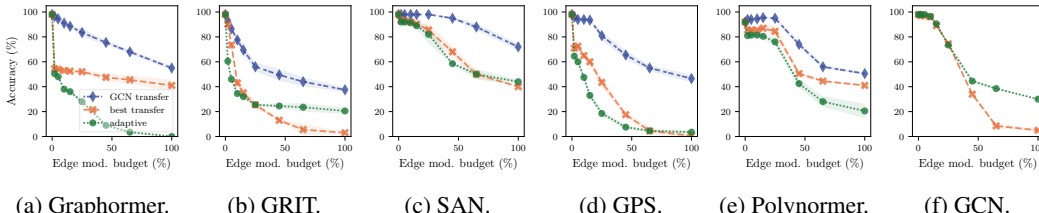

| (a) Graphormer. | (b) GRIT. | (c) SAN. | (d) GPS. | (e) Polynormer. | (f) GCN. |

Figure 7: Transfer attack results (node injection, evasion) for UPFD gossipcop (graph classification).

Table 1: Ablations for the Graphormer relaxations for a fixed budget of 1% for CLUSTER without and with perturbation constraints (c.), and 10% for UPFD politifact (pol.) and gossipcop (gos.). The mean and standard deviation over 4 runs with different seeds are reported.

| Deg. | SPD | Acc. (%) | | Node prob. | Acc. (%) | |
|---|---|---|---|---|---|---|
| | | CLUSTER | CLUSTER c. | | UPFD pol. | UPFD gos. |
| ✓ | ✓ | $52.61 \pm 0.57$ | $\mathbf{60.00} \pm 0.42$ | ✓ | $67.0 \pm 2.0$ | $\mathbf{38.0} \pm 0.0$ |
| ✓ | | $\mathbf{46.78} \pm 0.46$ | $68.45 \pm 0.37$ | ✓ | $67.0 \pm 2.0$ | $\mathbf{38.0} \pm 0.0$ |
| | ✓ | $50.81 \pm 0.41$ | $60.66 \pm 0.21$ | ✓ | $\mathbf{66.5} \pm 1.9$ | $39.5 \pm 1.9$ |
| | | | | ✓ | $\mathbf{66.5} \pm 1.9$ | $38.5 \pm 1.0$ |
| ✓ | ✓ | | | | $80.5 \pm 3.4$ | $53.5 \pm 1.0$ |
| random | | $66.52 \pm 0.61$ | $70.29 \pm 0.32$ | | $85.0 \pm 2.6$ | $61.5 \pm 4.1$ |
| clean | | $77.89$ | $77.89$ | | $92.0$ | $98.0$ |

GT models are more similar to each other than to a GCN. In some cases, *best transfer* is the overall strongest attack. However, note that choosing the best from up to eight (adaptively generated) attacks can be considered an ensemble with high computational cost. However, *best transfer* can be used as a "unit test" before laboriously designing adaptive attacks for a new GT architecture. Results of *best transfer* for other datasets and all individual transfer attacks are available in § E.2 & § E.3 respectively.

**Ablations**. We enable each of the continuous relaxations individually and together in different combinations. We report the results for Graphormer in Tab. 1. The node probability relaxation only applies to the node injection attacks on UPFD. The main insights from the results are: **(a)** All continuous relaxations individually seem to give somewhat useful gradients and can be used to get better results than the gradient-free random baseline. **(b)** For node injection attacks, using only the node probabilities in the graph pooling and to bias the attention scores is usually sufficient and leads to some of the strongest attack results. **(c)** Some relaxations are more effective than others, and using multiple does not seem to always work better than only one. However, one is not consistently better than the other. A good approach might be to try the relaxations individually, to find which are most relevant. Similar effects have been reported by (Tramèr et al., 2020) (Recurring Attack Theme T2). In § E.4, we also show ablations for GRIT and SAN components in Tab. 2 and Tab. 3 respectively, from which we can draw the same conclusions.

## 7 ADVERSARIAL TRAINING

We based our implementation of the adversarial training on the 'Free' adversarial training of Shafahi et al. (2019). The main idea is to couple the attack and training updates by 'replaying' the same mini-batch $k$ times for each of which both an attack and a training optimization step is made. This enables finding stronger perturbations whithout the large overhead of only performing a single model update for every $k$ attack steps of 'traditional' adversarial training. Note that we make some slight modifications to fit the limitations of our setting which are described in § F.

We evaluate our advarsarial training using Graphormer, which was one of the least robust models in our attack evaluation and thus has a large potential for robustness gains. As a baseline comparison we also include GCN. We evaluate node injection attacks on the UPFD datasets, the results for politifact are shown in Figs. 8a & 8b and for gossipcop in Figs. 8c & 8d. We show results for training configurations with different attack budgets and number of replay steps.

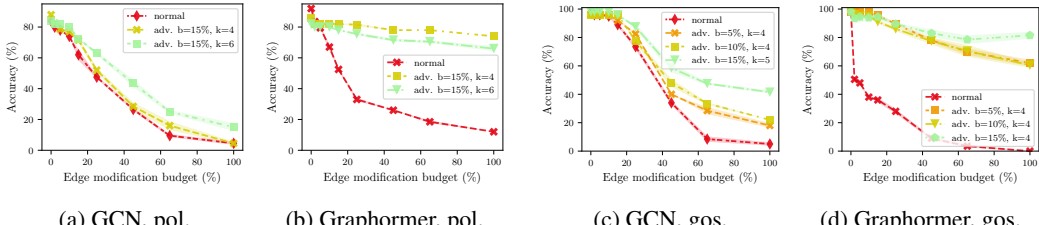

(a) GCN, pol.  (b) Graphormer, pol.  (c) GCN, gos.  (d) Graphormer, gos.

Figure 8: Node injection (evasion) attack results for adversarially trained models on UPFD politifact (pol) and gossipcop (gos) (graph classification).

On politifact the GCN adversarial training increases robustness a bit and even increases the clean accuracy. For Graphormer, there is a notable clean accuracy drop, however, the resulting models are very robust compared to both the normally trained Graphormer and the robust GCN models. For Gossipcop, adversarial training with GCN works quite well. The normally trained GCN was already very robust for small budgets, but the adversarial training makes it more robust for larger budgets as well. The normally trained Graphormer is catastrophically brittle. But with adversarial training it becomes remarkably robust, much more even than the GCN. Moreover, there is no major clean accuracy drop for neither model. These results indicate that the increased flexibility and capacity of graph transformers may be advantageous for learning robust models via adversarial training, even when the normally trained models are extremely non-robust.

## 8 RELATED WORK

Triggered by the seminal works of Zügner et al. (2018); Dai et al. (2018), a research area emerged spanning attacks, defenses, and certification of GNNs (Jin et al., 2021; Günnemann, 2022; Schuchardt et al., 2021; Scholten et al., 2022; Guerranti et al., 2023; Gosch et al., 2023b). However, GTs have been entirely neglected despite their success on common benchmarks. Zhu et al. (2024) is the sole exception acknowledging this gap. However, they solely propose a robust and sparse transformer and evaluate it with non-adaptive poisoning attacks. Thus, they do not shine light on the robustness of the diverse set of graph transformers nor do they study adaptive attacks.

Our attack is rooted in the GNN robustness literature. Xu et al. (2019) proposes the first Projected Gradient Descent attack for discrete L0 perturbations of the graph structure, with a focus on message-passing architectures. Geisler et al. (2021) extend this PGD with a randomization scheme to obtain the efficient Projected Randomized Block Coordinate (PRBCD) attack. Gosch et al. (2023a) extend PRBCD with local constraints, which is comparable with our relaxed GTs. While highlighting the general nature of our framework, we leave the empirical evaluation for future work. Further important related works are Lin et al. (2022); Zhu et al. (2018); Bojchevski & Günnemann (2019), where the authors study similar approximations for perturbations on the eigen-decomposition of the graph Laplacian. Moreover, Wang et al. (2023) attack message-passing architectures on the UPFD fake news detection using reinforcement learning. As an entry to Node Insertion Attacks (NIA), we refer to Wang et al. (2020); Zou et al. (2021).

## 9 CONCLUSION

We are the first to study the adversarial robustness of GTs and we provide the guiding principles for designing adaptive attacks. We study five representative GTs which use three of the most commonly used PEs: random-walk-based PEs; distance-based PEs; and spectral PEs. We empirically demonstrate that GTs can be catastrophically fragile in some settings and remarkably robust w.r.t. the studied attack in other settings. This diverse picture underlines the importance and need for adaptive attacks to reveal nuanced robustness properties. Similarly, also the comparison of GT's and GNN's robustness w.r.t. the studied attacks does not allow for a conclusion about which approach is superior in terms of robustness. Nevertheless, our work sets the important cornerstone for empirical research in answering this very question. Finally, we leverage our adaptive attacks to obtain the first results for adversarial training with GTs. These results are promising and show that GTs have the potential to become very robust against graph structure perturbations.

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

## A  LIMITATIONS

All our adversarial attacks do neither provide a guarantee nor insights about how close they are to the optimal adversarial example. We note that this is common practice in the adversarial learning community. Nevertheless, if we are able to find an adversarial example, this proves the non-robustness of the studied model. We conducted the experiment with utmost care, s.t. we expect the results to be reasonable upper bounds that will be challenging to beat for future work. We still state clearly that due to the high computational cost of dense GTs, we cannot certify that these efforts are sufficient (e.g., our hyperparameter searches were comprehensive yet not exhaustive).

## B  BROADER IMPACT

While the threat model of attacking fake news detection could have a negative societal impact, our methods are applicable mostly in a white-box setting and, therefore, are much more useful to those who are developing fake news detection to probe and improve the robustness of their models. If a model developer has access to the right tools, we expect its information advantage to outweigh the availability of attacks by far.

## C  GRAPH TRANSFORMER ARCHITECTURE DETAILS

**Random-walk-based GRIT**. The GRIT model architecture (Ma et al., 2023) terms their PE Relative Random Walk Probabilities (RRWP). The random walk encodings are collected from a fixed-length walk of length $k$, which is a hyperparameter (usually $k > 10$). The PEs are based on the tensor:

$$\boldsymbol{P} = [\boldsymbol{I}, \boldsymbol{M}, \boldsymbol{M}^2, ..., \boldsymbol{M}^{k-1}] \in \mathbb{R}^{n \times n \times k}, \quad \text{with} \quad \boldsymbol{M} = \boldsymbol{D}^{-1}\boldsymbol{A} \tag{14}$$

This yields an embedding vector $\boldsymbol{P}_{ij} \in \mathbb{R}^k$ for each of the $n^2$ node-pairs $(v_i, v_j)$. The diagonal vector entries are transformed to dimension $d$ by a linear layer and added to the node features as PEs: $\boldsymbol{h}_i^{(0)} = \boldsymbol{x}_i + g_1(\boldsymbol{P}_{ii})$. Additionally, all $n^2$ vectors are transformed by a separate linear layer and added as node-pair features: $\boldsymbol{h}_{i,j}^{(0)} = g_2(\boldsymbol{P}_{ij})$. The node representations $\boldsymbol{h}_i$ and node-pair representations $\boldsymbol{h}_{i,j}$ are updated in each transformer layer by a modified attention mechanism, which includes an adaptive degree-scaler that is applied to the node representations:

$$\tilde{\boldsymbol{h}}_i = (\boldsymbol{h}_i \odot \boldsymbol{\theta}_1) + \log(1 + \deg(v_i)) \cdot (\boldsymbol{h}_i \odot \boldsymbol{\theta}_2) \tag{15}$$

where $\boldsymbol{\theta}_1, \boldsymbol{\theta}_2 \in \mathbb{R}^d$ are learnable weights.

**Distance-based Graphormer**. The Graphormer model (Ying et al., 2021) uses degree PEs. For each discrete degree value there is a learnable embedding vector $\boldsymbol{z} \in \mathbb{R}^d$. The embeddings are added to the node features according to the node degrees:

$$\boldsymbol{h}_i^{(0)} = \boldsymbol{x}_i + \boldsymbol{z}_{\deg(v_i)} \tag{16}$$

Similarly, a learnable scalar $b \in \mathbb{R}$ is assigned to each discrete Shortest Path Distance (SPD). This value is added to the raw attention scores and results in a re-weighting of the attention weights after applying the softmax function:

$$\hat{w}_{ij} = w_{ij} + b_{\mathsf{spd}(v_i, v_j)}, \qquad \alpha_{ij} = \mathsf{softmax}(\hat{w}_{ij}) \tag{17}$$

where $w_{ij} = \boldsymbol{W}_q \boldsymbol{h}_i \cdot \boldsymbol{W}_k \boldsymbol{h}_j / \sqrt{d}$. For graph-level tasks, a virtual node is added to the graph with its own distinct learnable bias $b_{\mathsf{virtual}}$, which is used as graph representation in the pooling stage.

**Spectral SAN**. The SAN architecture (Kreuzer et al., 2021) uses learned Laplacian-based PEs (LPEs), starting with the eigen-decomposition of the Laplacian $\boldsymbol{L}_{sym} = \boldsymbol{U}\boldsymbol{\Lambda}\boldsymbol{U}^{\mathsf{T}}$, where diagonal entries of $\boldsymbol{\Lambda}_{ii} = \lambda_i$ are the eigenvalues of $\boldsymbol{L}_{sym}$ in ascending order $\lambda_1 \leq \lambda_2 \leq ... \leq \lambda_n$, and the columns of $\boldsymbol{U}$ are the corresponding eigenvectors. Determined by a hyperparameter, only the $k$-th smallest eigenvalues and their eigenvectors are used, which we write as $\boldsymbol{\Lambda}_k \in \mathbb{R}^{k \times k}$ and $\boldsymbol{U}_k \in \mathbb{R}^{n \times k}$. For each node $v_i$, its PEs are initialized as the concatenation of the eigenvalues and the $i$-th row of $\boldsymbol{U}_k$:

$$\boldsymbol{P}_i = [\mathsf{diag}(\boldsymbol{\Lambda}_k) \,\|\, (\boldsymbol{U}_k)_i] \in \mathbb{R}^{k \times 2} \tag{18}$$

Further processing by a transformer encoder results in $\boldsymbol{p}_i = f(\boldsymbol{P}_i) \in \mathbb{R}^{d_p}$, which is concatenated to the node features: $\boldsymbol{h}_i^{(0)} = \boldsymbol{x}_i \parallel \boldsymbol{p}_i$. Additionally, the main graph transformer attention mechanism is modified to have two separate key and query weights for connected and unconnected node-pairs. The attention scores to the connected nodes and to the unconnected nodes are computed independently, each with a softmax. A hyperparameter $\gamma \in \mathbb{R}^+$ controls how the two scores are relatively scaled, varying the bias towards sparse or full attention:

$$\alpha_{ij} = \begin{cases} \frac{1}{1+\gamma} \mathsf{softmax} \left( \boldsymbol{W}_{q,\mathrm{real}} \boldsymbol{h}_i \cdot \boldsymbol{W}_{k,\mathrm{real}} \boldsymbol{h}_j / \sqrt{d} \right) & \text{if } (v_i, v_j) \text{ is a real edge} \\ \frac{\gamma}{1+\gamma} \mathsf{softmax} \left( \boldsymbol{W}_{q,\mathrm{fake}} \boldsymbol{h}_i \cdot \boldsymbol{W}_{k,\mathrm{fake}} \boldsymbol{h}_j / \sqrt{d} \right) & \text{otherwise} \end{cases} \tag{19}$$

## D  NODE INJECTION ATTACK DETAILS

Let $\mathcal{D} = \{\mathcal{G}_1, ..., \mathcal{G}_N\}$ be the dataset including all graphs, where each graph $\mathcal{G}_i = (\mathcal{V}_i, \mathcal{E}_i)$ has $n_i$ nodes $V_i = \{v_{i,1}, ..., v_{i,n_i}\}$ with node feature matrix $\boldsymbol{X}_i \in \mathbb{R}^{n_i \times d}$. The total number of nodes in the dataset is $n_{\mathcal{D}} = \sum n_i$. Let $\mathcal{G}_{atk}$ be the graph that is being attacked. We define the candidate set of injection nodes as the union of the nodes of all other graphs: $\mathcal{V}_{cs} = \bigcup_{\mathcal{G}_i \in \mathcal{D} \setminus \mathcal{G}_{atk}} \mathcal{V}_i$, which includes $n_{cs} = n_{\mathcal{D}} - n_{atk}$ nodes with corresponding features $\boldsymbol{X}_{cs}$. It is of course possible to restrict this candidate set if is is not sensible or not feasible to include all nodes.

We can augment the original (connected) graph $\mathcal{G}_{atk} = (\boldsymbol{A}_{atk}, \boldsymbol{X}_{atk})$ by adding the injection candidate set as isolated nodes:

$$\mathcal{G}'_{atk} = (\boldsymbol{A}'_{atk}, \boldsymbol{X}'_{atk}), \quad \boldsymbol{A}'_{atk} = \begin{bmatrix} \boldsymbol{A}_{atk} & \boldsymbol{0} \\ \boldsymbol{0} & \boldsymbol{0} \end{bmatrix} \in \{0,1\}^{n_{\mathcal{D}} \times n_{\mathcal{D}}}, \quad \boldsymbol{X}'_{atk} = \begin{bmatrix} \boldsymbol{X}_{atk} \\ \boldsymbol{X}_{cs} \end{bmatrix} \in \mathbb{R}^{n_{\mathcal{D}} \times d} \tag{20}$$

Edge-flip perturbations to this augmented adjacency matrix, $\tilde{\boldsymbol{A}}' = \boldsymbol{A}' + \delta \boldsymbol{A}'$, model both structure perturbations and node injections together. As in Eq. 2, the perturbation $\delta \boldsymbol{A}'$ can be expressed in terms of a binary edge flip matrix: $\tilde{\boldsymbol{A}}' = \boldsymbol{A}' + (\boldsymbol{1}_n \boldsymbol{1}_n^{\mathsf{T}} - 2\boldsymbol{A}') \odot \boldsymbol{B}'$, where:

$$\boldsymbol{B}' = \begin{bmatrix} \boldsymbol{B} & \boldsymbol{E} \\ \boldsymbol{E}^{\mathsf{T}} & \boldsymbol{F} \end{bmatrix} \in \{0,1\}^{n_{\mathcal{D}} \times n_{\mathcal{D}}} \tag{21}$$

Since the number of edge flips is bounded by a budget usually much smaller than the candidate set size i.e., $\Delta \ll n_{cs}$, the perturbed augmented graph $\tilde{\mathcal{G}} = (\tilde{\boldsymbol{A}}', \boldsymbol{X}')$ still mostly contains isolated nodes. Therefore, we prune away all disconnected components, which for the unperturbed graph simply reverts the augmentation: $\mathsf{prune}(\mathcal{G}') = \mathcal{G}$. However, for a perturbed augmented graph, this results in the perturbed graph that we are seeking:

$$\tilde{\mathcal{G}} = \mathsf{prune}(\tilde{\boldsymbol{A}}', \boldsymbol{X}') = (\tilde{\boldsymbol{A}}, \tilde{\boldsymbol{X}}), \quad \tilde{\boldsymbol{A}} \in \{0,1\}^{\tilde{n} \times \tilde{n}}, \quad \tilde{\boldsymbol{X}} \in \mathbb{R}^{\tilde{n} \times d} \tag{22}$$

Here, $n_{in}$ is the number of injected nodes, and $\tilde{n} = n + n_{in}$ is the total number of nodes of the perturbed graph. The NIA objective can thus be written as:

$$\max_{\boldsymbol{B}' \text{ s.t. } ||\boldsymbol{B}'||_0 < \Delta} \mathcal{L}_{atk}(f_\theta(\tilde{\mathcal{G}})), \quad \text{with} \quad \tilde{\mathcal{G}} = \mathsf{prune}(\boldsymbol{A}' + (\boldsymbol{1}_n \boldsymbol{1}_n^{\mathsf{T}} - 2\boldsymbol{A}') \odot \boldsymbol{B}', \tilde{\boldsymbol{X}}') \tag{23}$$

where, $f_\theta$ is the trained GNN and $\mathcal{L}_{atk}$ is a suitable attack loss. Note that the edge flip budget $\Delta$ is also an upper bound for the number nodes that can be injected: $0 \le n_{in} \le \Delta$.

**Edge block sampling**. To optimize the objective, we can apply the relaxation $\boldsymbol{B}_{ij} \in [0, 1]$, as shown in § 2.1. In this case, PRBCD (Geisler et al., 2021) not only enables more efficient optimization, but setting a smaller block size is crucial to limit the number of connected injection nodes during optimization, since GTs complexity scales with $O(\tilde{n}^2)$. Moreover, the edge sampling allows us to control which parts of $\boldsymbol{B}'$ in Eq. 21 to sample from, e.g. not sampling in $\boldsymbol{B}$ results in 'pure' node injections without modifying edges in the original graph. For NIAs with large candidate sets, we only sample from $\boldsymbol{E}$, as sampling from the $n_{cs}^2$ entries of $\boldsymbol{F}$ results in many disconnected injection node pairs that get pruned away.

### D.1  NODE PROBABILITY EXAMPLE

We provide an illustrative example in Fig. 9 of how the iterative node probability is applied. Each iteration of Eq. 12 can be thought of as a message passing step to update the node probability

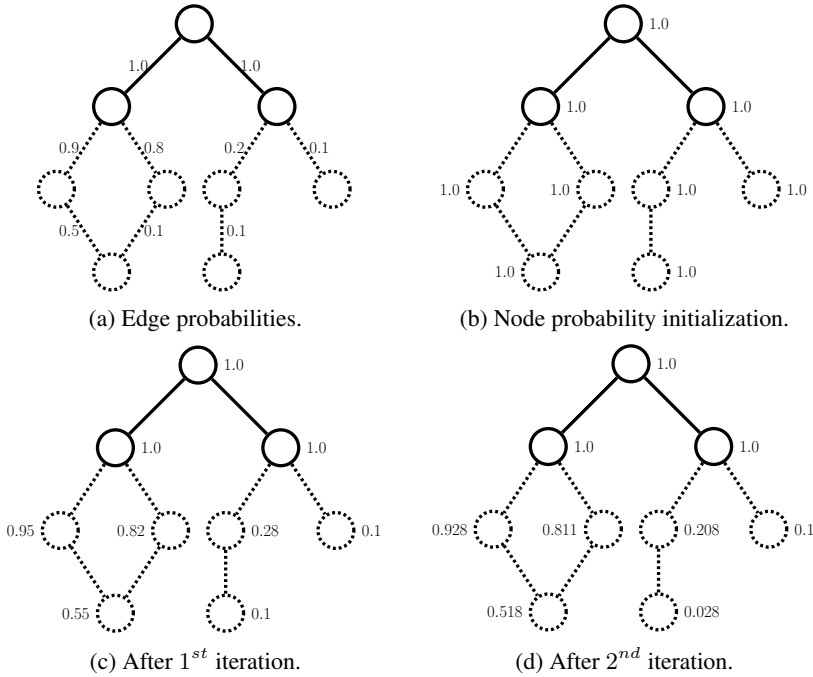

(a) Edge probabilities.      (b) Node probability initialization.

(c) After $1^{st}$ iteration.      (d) After $2^{nd}$ iteration.

Figure 9: Node probability example. Dashed lines indicate injection nodes.

approximation based on the neighbors current approximations:

$$p_i^{(t+1)} = 1 - \prod_{v_j \in \mathcal{N}_i} (1 - \boldsymbol{A}_{ij} \cdot p_j^{(t)}), \quad \text{with} \quad p_i^{(0)} = 1$$

The number of iterations should be set in the order of expected longest chain of added injection nodes. Therefore very few iterations (2-5) should suffice for most NIAs.

# E   ADDITIONAL ATTACK RESULTS

## E.1   CLUSTER CONSTRAINED ATTACK

Fig. 10 shows the attack results for the CLUSTER dataset when constraining edge perturbations such that edges to the labeled nodes cannot be flipped. As expected, this significantly reduces the attack strength compared to the unconstrained setting shown in Fig 3.

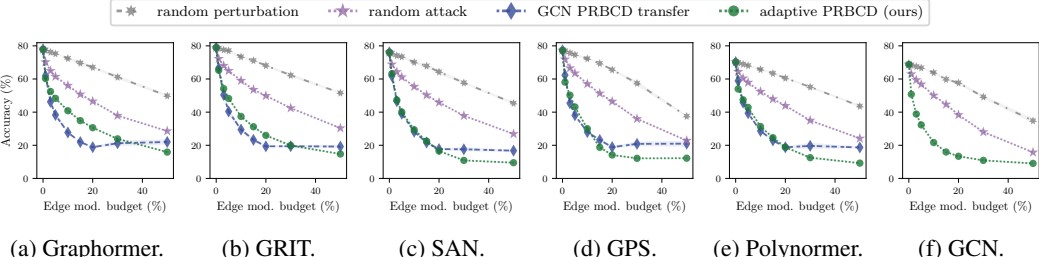

(a) Graphormer.    (b) GRIT.    (c) SAN.    (d) GPS.    (e) Polynormer.    (f) GCN.

Figure 10: CLUSTER constrained attack results.

### E.2 BEST TRANSFER ATTACKS

Here we provide the results for the best transfer results, analogous to Fig. 7 but for all additional datasets. Results for CLUSTER are in Fig. 11, for CLUSTER (constrained) in Fig. 12, for Reddit Threads in Fig. 13, and for UPFD politifact in Fig. 14.

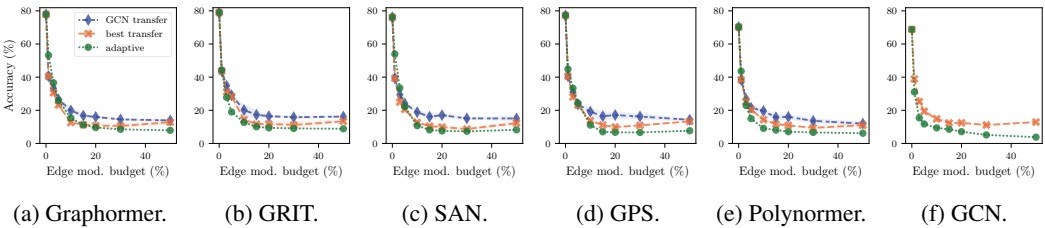

(a) Graphormer.  (b) GRIT.  (c) SAN.  (d) GPS.  (e) Polynormer.  (f) GCN.

Figure 11: Best transfer, CLUSTER (inductive node classification), structure attack (global, evasion).

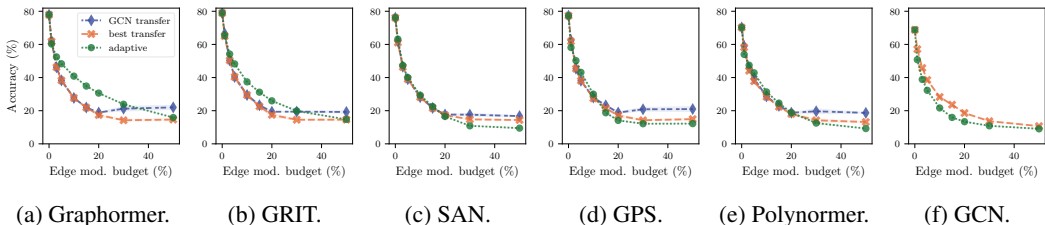

(a) Graphormer.  (b) GRIT.  (c) SAN.  (d) GPS.  (e) Polynormer.  (f) GCN.

Figure 12: Best transfer, CLUSTER (inductive node classification), constrained structure attack (global, evasion).

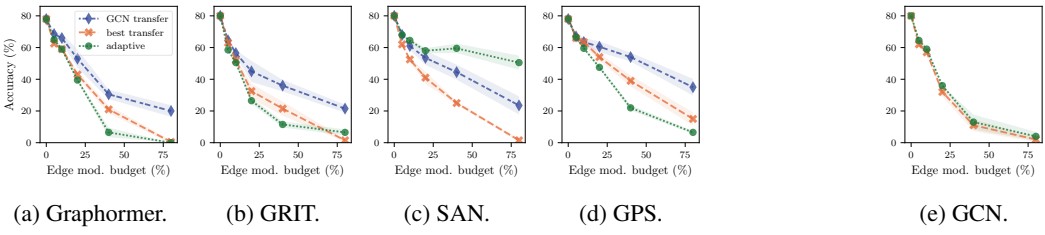

(a) Graphormer.  (b) GRIT.  (c) SAN.  (d) GPS.     (e) GCN.

Figure 13: Best transfer, Reddit Threads (graph classification), structure attack (evasion).

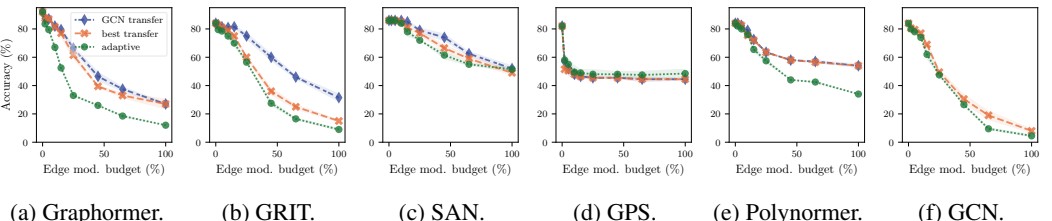

(a) Graphormer.  (b) GRIT.  (c) SAN.  (d) GPS.  (e) Polynormer.  (f) GCN.

Figure 14: Best transfer, UPFD politifact (graph classification), node injection attack (evasion).

### E.3 ALL TRANSFER ATTACKS

Here we provide the more detailed (but less readable) attack results including the individual transfer models. Results for Graphormer are in Fig. 15, for GRIT in Fig. 16, for SAN in Fig. 17, for GPS in Fig. 18, for Polynormer in Fig. 19, and for GCN in Fig. 20.

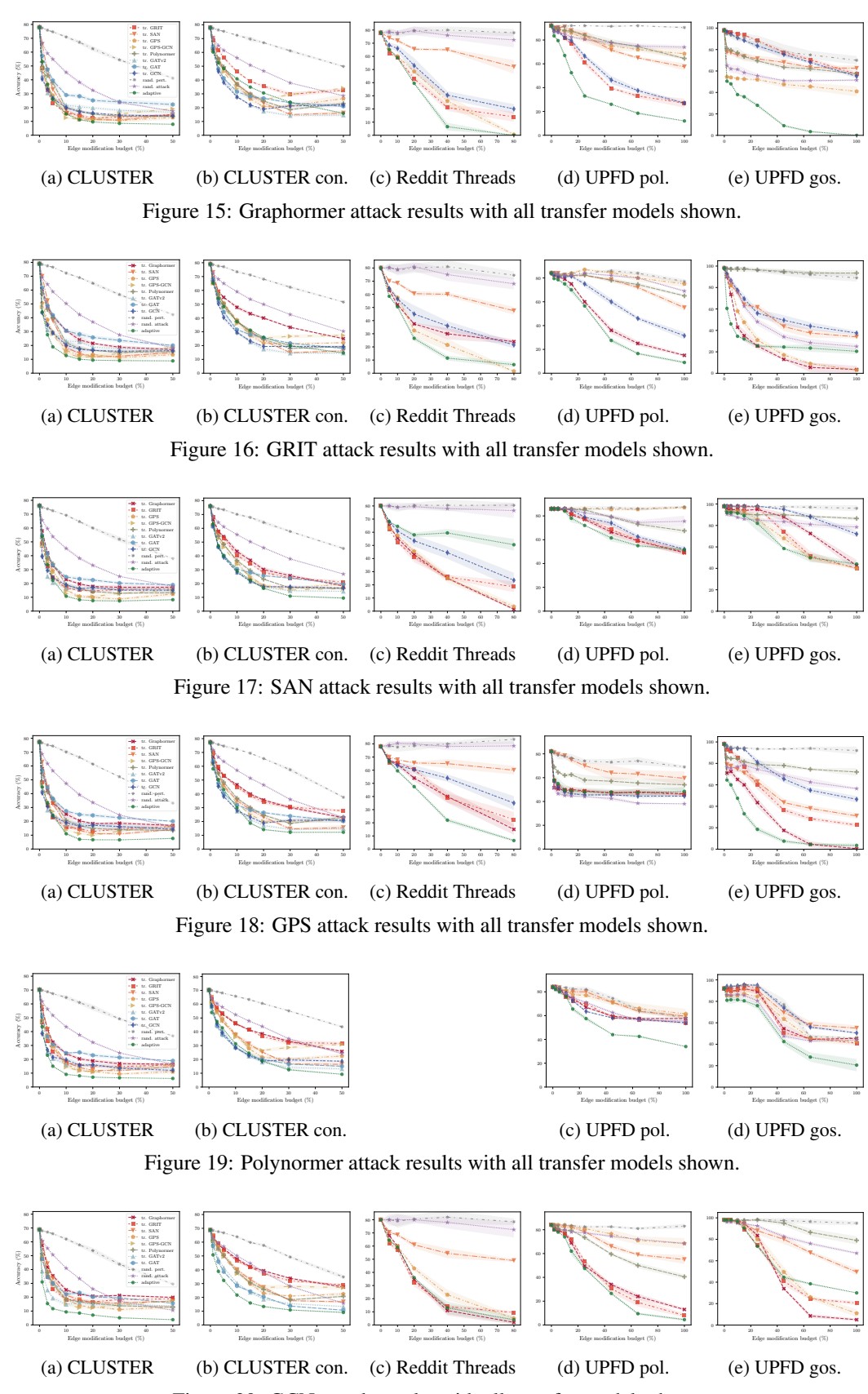

Figure 15: Graphormer attack results with all transfer models shown.

Figure 16: GRIT attack results with all transfer models shown.

Figure 17: SAN attack results with all transfer models shown.

Figure 18: GPS attack results with all transfer models shown.

Figure 19: Polynormer attack results with all transfer models shown.

Figure 20: GCN attack results with all transfer models shown.

### E.4 ABLATIONS FOR GRIT AND SAN

We also check the attack strength for GRIT when enabling or disabling gradient computation through certain parts of the model and show the results in Tab. 2. It is possible to get strong attacks even without computing gradients through RRWP, which could be much more efficient computationally, depending on the model and graph size. For node injection attacks, as for the other models, using only the node probability bias in the attention scores already leads to the strongest attacks we report.

Ablation on different attack components on SAN are presented in Tab. 3. The colomn 'Eig. backp.' refers to the alternative method of obtaining gradients through the eigen-decomposition discussed in § H.2. The results indicate that both methods seem to work equally well.

Table 2: Ablations for the GRIT relaxations for a fixed budget of 1% for CLUSTER without and with perturbation constraints (c.), and 10% for UPFD politifact (pol.) and gossipcop (gos.). The mean and standard deviation over 4 runs with different seeds are reported.

| PE grad. | Deg. grad. | Acc. (%) | | Node prob. | Acc. (%) | |
|---|---|---|---|---|---|---|
| | | CLUSTER | CLUSTER c. | | UPFD pol. | UPFD gos. |
| ✓ | ✓ | **44.07** ± 0.79 | **65.25** ± 0.22 | ✓ | **34.5** ± 1.0 | 75.0 ± 2.6 |
| ✓ | | 46.27 ± 0.36 | 65.70 ± 0.35 | ✓ | **34.5** ± 1.0 | 74.5 ± 2.5 |
| | ✓ | 49.51 ± 0.90 | 66.49 ± 0.49 | ✓ | **34.5** ± 1.0 | **73.5** ± 1.0 |
| | | | | ✓ | **34.5** ± 1.0 | **73.5** ± 1.0 |
| ✓ | ✓ | | | | 54.5 ± 1.9 | 83.0 ± 2.0 |
| random | | 69.13 ± 0.10 | 72.25 ± 0.29 | | 76.0 ± 4.3 | 82.0 ± 0.0 |
| clean | | 78.98 | 78.98 | | 98.0 | 84.0 |

Table 3: Ablations for the SAN relaxations for a fixed budget of 1% for CLUSTER without and with perturbation constraints (c.), and 10% for UPFD politifact (pol.) and gossipcop (gos.). The mean and standard deviation over 4 runs with different seeds are reported.

| Attn. | Lap. pert. | Eig. backp. | Acc. (%) | | Node prob. | Acc. (%) | |
|---|---|---|---|---|---|---|---|
| | | | CLUSTER | CLUSTER c. | | UPFD pol. | UPFD gos. |
| ✓ | ✓ | | 54.0 ± 0.6 | 63.3 ± 0.3 | ✓ | 83.5 ± 1.0 | 91.5 ± 4.1 |
| ✓ | | ✓ | 54.4 ± 0.6 | **62.9** ± 0.2 | ✓ | 82.0 ± 2.3 | 94.0 ± 3.0 |
| ✓ | | | **53.9** ± 0.3 | 63.2 ± 0.1 | ✓ | 77.5 ± 1.0 | 91.5 ± 2.5 |
| | ✓ | | 57.1 ± 0.6 | 67.2 ± 0.2 | ✓ | 83.5 ± 1.0 | 89.5 ± 1.9 |
| | | ✓ | 55.1 ± 1.0 | 67.3 ± 0.3 | ✓ | 81.0 ± 1.2 | 89.5 ± 3.4 |
| | | | | | ✓ | **77.0** ± 1.2 | 89.5 ± 3.4 |
| ✓ | ✓ | | | | | 86.0 ± 0.0 | 90.1 ± 6.0 |
| ✓ | | ✓ | | | | 86.0 ± 2.3 | 91.0 ± 5.3 |
| random | | | 65.7 ± 0.7 | 68.9 ± 0.3 | | 86.0 ± 0.0 | **87.5** ± 1.0 |
| clean | | | 76.1 | 76.1 | | 86.0 | 98.0 |

# F   ADVERSARIAL TRAINING ALGORITHM

We based our implementation of the adversarial training on the 'Free' adversarial training of Shafahi et al. (2019). Pseudocode for our adversarial training is given in Alg. 1. The main modifications are that we do two separate forward and backwards passes for the attack and model respectively. This is because: (1) the attack and model often have distinct loss functions that they optimize for, and (2) we sample a discrete structure perturbation for the model, such that the perturbed graph is included in the original valid sample space. Another difference is that we need to iterate over the graphs in the minibatch separately. This is a limitation caused by: (1) The attack optimization steps are not trivial to parallelize, especially for node injection attacks, and (2) the PE computations (e.g. Laplacian eigen-decomposition) are also not easily parallelizable and need to re-computed for each new perturbed graph.

Given these limitations, our adversarial training is much less efficient. It requires at least $2 \cdot |\mathcal{B}|$ times more model evaluations than normal training. Furthermore, for many GTs the PE computation is one of the most computationally expensive steps. Therefore, PEs are usually precomputed in a pre-processing step. During adversarial training, we need to compute PEs for new unseen perturbations at each step, which further increases the overhead. Nonetheless, following the main idea of Shafahi et al. (2019) alleviates some of the overhead and makes it somewhat practically feasible.

---

**Algorithm 1** Our $k$-step 'free' adversarial training

---

**Require:** Training dataset $\mathcal{T}$, model $f_\theta$, attack budget $\Delta$, number of steps $k$, learning rate $\alpha$

    Initialize $\theta$

    **for** epoch = $1...N_{ep}/k$ **do**

        **for** minibatch $\mathcal{B} \subset \mathcal{T}$ **do**

            Initialize perturbations $\boldsymbol{P}$

            **for** $i = 1...k$ **do**

                $g_\theta \leftarrow 0$

                **for** graph $\mathcal{G} = (\boldsymbol{A}, \boldsymbol{X}, \boldsymbol{y}) \in \mathcal{B}$ **do**

                    $\boldsymbol{P} \leftarrow \text{PRBCD\_step}(f_\theta, \boldsymbol{X}, \boldsymbol{A}, \boldsymbol{P}, \Delta)$

                    $\boldsymbol{A}' \leftarrow \text{sample\_discrete}(\boldsymbol{A}, \boldsymbol{P})$

                    $g_\theta \leftarrow g_\theta \nabla_\theta \mathcal{L}(f_\theta(\boldsymbol{A}', \boldsymbol{X}), \boldsymbol{y})$

                **end for**

                $\theta \leftarrow \theta + \alpha \cdot \frac{1}{|\mathcal{B}|} \cdot g_\theta$

            **end for**

        **end for**

    **end for**

---

# G   HYPERPARAMETERS

We include the hyperparameters for Graphormer in Tab. 4, for GRIT in Tab. 5, for SAN in Tab. 6, and for GCN in Tab. 7.

Table 4: Hyperparameters for Graphormer.

|  | CLUSTER | UPFD pol. | UPFD gos. |
|---|---|---|---|
| Optimizer | Adam | AdamW | AdamW |
| Learning rate | $8.04 \times 10^{-4}$ | $1.29 \times 10^{-4}$ | $3.75 \times 10^{-4}$ |
| Weight decay | - | $6.7 \times 10^{-3}$ | $3.75 \times 10^{-4}$ |
| Max. deg. | 70 | 37 | 21 |
| Max. dist. | 4 | 10 | 8 |
| Attention dropout | 0.107 | 0.0 | 0.382 |
| Input dropout | 0.0 | 0.0 | $8.65 \times 10^{-3}$ |
| Dropout | 0.069 | 0.0 | 0.069 |
| Hidden dimension | 60 | 40 | 30 |
| Layers | 15 | 6 | 8 |
| Attention heads | 6 | 8 | 3 |
| Graph pooling | - | virtual node | virtual node |

Table 5: Hyperparameters for GRIT.

|  | CLUSTER | UPFD pol. | UPFD gos. |
|---|---|---|---|
| Optimizer | AdamW | AdamW | AdamW |
| Learning rate | $1.29 \times 10^{-3}$ | $5.61 \times 10^{-4}$ | $2.24 \times 10^{-3}$ |
| Weight decay | $4.16 \times 10^{-6}$ | $2.97 \times 10^{-2}$ | $1.20 \times 10^{-8}$ |
| RRWP max. steps | 4 | 9 | 6 |
| Attention dropout | 0.478 | 0.490 | 0.292 |
| Dropout | 0.100 | 0.0 | 0.056 |
| Hidden dimension | 48 | 9 | 18 |
| Layers | 12 | 2 | 6 |
| Attention heads | 8 | 3 | 6 |
| Graph pooling | - | add | add |

Table 6: Hyperparameters for SAN.

|  | CLUSTER | UPFD pol. | UPFD gos. |
|---|---|---|---|
| Optimizer | Adam | Adam | Adam |
| Learning rate | $5.0 \times 10^{-4}$ | $1.29 \times 10^{-4}$ | $5.41 \times 10^{-4}$ |
| Weight decay | 0.0 | $1.0 \times 10^{-3}$ | 0.0 |
| Max. eig. | 10 | 10 | 24 |
| PE dim. | 16 | 16 | 20 |
| PE layers | 1 | 2 | 2 |
| PE heads | 4 | 4 | 5 |
| gamma | 0.1 | $1.43 \times 10^{-2}$ | $4.28 \times 10^{-3}$ |
| Dropout | 0.0 | 0.0 | $1.73 \times 10^{-2}$ |
| Hidden dimension | 48 | 96 | 80 |
| Layers | 16 | 3 | 3 |
| Attention heads | 8 | 4 | 8 |
| Graph pooling | - | add | add |

Table 7: Hyperparameters for GCN.

|  | CLUSTER | UPFD pol. | UPFD gos. |
|---|---|---|---|
| Optimizer | Adam | AdamW | AdamW |
| Learning rate | $1.00 \times 10^{-3}$ | $5.29 \times 10^{-3}$ | $1.23 \times 10^{-4}$ |
| Weight decay | - | $2.59 \times 10^{-2}$ | 2.85 |
| Dropout | 0.0 | 0.0 | 0.5 |
| Hidden dimension | 172 | 473 | 105 |
| Layers | 16 | 2 | 3 |
| Graph pooling | - | add | add |

## H LAPLACIAN EIGEN-DECOMPOSITION GRADIENT

### H.1 PERTURBATION APPROXIMATION: REPEATED EIGENVALUES

Unfortunately, Eq. 10 and 11 do not hold in general when repeated eigenvalues are present. This is due to the fact that a small perturbation can separate repeated eigenvalues into distinct eigenvalues. For the unperturbed graph, the choice of eigenvector basis of the repeated eigenvalue's eigenspace is arbitrary. In the perturbed graph, however, the eigenvectors corresponding to the now distinct eigenvalues are uniquely defined (up to the sign). Thus, a large discontinuous change in the eigenvectors can be caused by an arbitrarily small input perturbation. For instance, consider the matrix $M$ with repeated eigenvalue 1 and the following valid eigendecomposition:

$$M = \begin{bmatrix} 1 & 0 \\ 0 & 1 \end{bmatrix} = U \Lambda U^{\mathsf{T}}, \quad \Lambda = \begin{bmatrix} 1 & 0 \\ 0 & 1 \end{bmatrix}, \quad U = \frac{\sqrt{2}}{2} \begin{bmatrix} 1 & 1 \\ 1 & -1 \end{bmatrix} \tag{24}$$

As soon as an arbitrarily small perturbation $\varepsilon$ is added to one of the diagonal entries, the eigenvalues become distinct and the choice of eigenvectors becomes constrained, which results in a discontinuous change:

$$\tilde{M} = \begin{bmatrix} 1 & 0 \\ 0 & 1 + \varepsilon \end{bmatrix} = \tilde{U} \tilde{\Lambda} \tilde{U}^{\mathsf{T}}, \quad \tilde{\Lambda} = \begin{bmatrix} 1 & 0 \\ 0 & 1 + \varepsilon \end{bmatrix}, \quad \tilde{U} = \begin{bmatrix} 1 & 0 \\ 0 & 1 \end{bmatrix} \tag{25}$$

However, there is always some valid choice of eigenvectors in the unperturbed graph that leads to a continuous change with respect to the given perturbation, e.g., in the above example $\tilde{U}$ is also a valid choice for the eigenvectors $U$ of the unperturbed matrix. With the right choice of unperturbed eigenvectors, the approximation equations are, therefore, still valid. Here, we provide a procedure to transform arbitrary eigenvectors into the ones that lead to good perturbation approximations. For the theory showing why this leads to the correct result, we refer to Bamieh (2022).

Let $(\Lambda, \hat{U})$ be the output of the eigendecomposition algorithm for the unperturbed Laplacian $L_{sym}$ containing repeated eigenvalues. We can write the eigendecomposition in it's block form:

$$L_{sym} = \hat{U} \Lambda \hat{U}^{\mathsf{T}}, \quad \Lambda = \begin{bmatrix} \Lambda_1 & & \\ & \ddots & \\ & & \Lambda_{n'} \end{bmatrix}, \quad \hat{U} = \begin{bmatrix} | & & | \\ \hat{U}_1 & \cdots & \hat{U}_{n'} \\ | & & | \end{bmatrix} \tag{26}$$

For a simple eigenvalue $\lambda_i$, the block has dimension one, i.e., $\Lambda_i = [\lambda_i]$ and $\hat{U}_i = u_i$. For a repeated eigenvalue $\lambda_j$ with multiplicity $r$, it's corresponding block is $\lambda_j I_r$ and $\hat{U}_j \in \mathbb{R}^{n \times r}$. Let $P = P^{\mathsf{T}}$ be an arbitrary symmetric perturbation to the original symmetric Laplacian. We can transform each eigenspace basis of a repeated eigenvalue $\hat{U}_j$ to the correct choice of eigenvectors as follows:

$$\begin{aligned} U_j &= \hat{U}_j Q \\ P_{\hat{U},j} &= \hat{U}_j^{\mathsf{T}} P \hat{U}_j = Q \Lambda_P Q^{\mathsf{T}} \in \mathbb{R}^{r \times r} \end{aligned} \tag{27}$$

First, we do a basis transformation of the perturbation matrix onto the eigenbasis $\hat{U}$. Then we find the eigendecomposition of the corresponding diagonal block $P_{\hat{U},j}$ and use these perturbation eigenvectors to transform the original Laplacian eigenvectors. This results in a choice of valid eigenvectors $U_j$ such that the approximations in Eq. 10 and 11 are valid for repeated eigenvalues and guarantees continuity of the eigenvalues and vectors with respect to a single perturbation, e.g., when linearly interpolating from the unperturbed to the fully perturbed matrix.

### H.2 BACKPROPAGATION: BREAKING UP REPEATED EIGENVALUES

The only thing preventing the use of auto-differentiation to compute gradients through the eigen-decomposition is the presence of repeated eigenvalues. As a workaround, Lin et al. (2022) propose adding small amplitude random noise to the entire adjacency matrix. While this usually separates the repeated eigenvalues, it is not guaranteed to. We propose a different approach in which the smallest possible perturbation term is added to the Laplacian matrix, such that the repeated eigenvalues are guaranteed to be separated while the eigenvectors remain unchanged.

To achieve this, we must first define a minimum eigenvalue distance hyperparameter $\varepsilon$, which we set to $10^{-4}$ in our experiments. Then we define eigenvalue separation such that for all perturbed Laplacian eigenvalues $|\hat{\lambda}_i - \hat{\lambda}_j| \geq \varepsilon$ must hold. Furthermore, we can define a vector $\boldsymbol{o} \in \mathbb{R}^n$ such that each entry represents the offset of the perturbed eigenvalue in relation to the true value:

$$\hat{\boldsymbol{\Lambda}} = \begin{bmatrix} \lambda_1 + \boldsymbol{o}_1 & & \\ & \ddots & \\ & & \lambda_n + \boldsymbol{o}_n \end{bmatrix} = \boldsymbol{\Lambda} + \text{diag}(\boldsymbol{o}) \tag{28}$$

In order for the perturbed matrix to have the same eigenvectors as the unperturbed Laplacian, we can define it by its eigendecomposition:

$$\begin{aligned} \hat{\boldsymbol{L}}_{sym} &= \boldsymbol{U}\hat{\boldsymbol{\Lambda}}\boldsymbol{U}^{\mathsf{T}} \\ &= \boldsymbol{U}(\boldsymbol{\Lambda} + \text{diag}(\boldsymbol{o}))\boldsymbol{U}^{\mathsf{T}} \\ &= \boldsymbol{U}\boldsymbol{\Lambda}\boldsymbol{U}^{\mathsf{T}} + \boldsymbol{U}\text{diag}(\boldsymbol{o})\boldsymbol{U}^{\mathsf{T}} \\ &= \boldsymbol{L}_{sym} + \boldsymbol{U}\text{diag}(\boldsymbol{o})\boldsymbol{U}^{\mathsf{T}} \end{aligned} \tag{29}$$

Consequently, the additive perturbation has the form $\boldsymbol{P} = \boldsymbol{U}\text{diag}(\boldsymbol{o})\boldsymbol{U}^{\mathsf{T}}$, such that it shares the same eigenvectors as the original Laplacian, and its eigenvalues are exactly the offsets.

Since the Frobenius norm can also be computed using the singular values, finding the perturbation with minimum norm is equivalent to minimizing the Euclidean norm of the offset vector $\|\boldsymbol{P}\|_F = \sqrt{\sum_i \boldsymbol{o}_i^2} = \|\boldsymbol{o}\|_2$. To ensure that the order of the eigenvalues is not changed we can define the separation constraints for the consecutive pairs of the perturbed eigenvalues $\hat{\lambda}_{i+1} - \hat{\lambda}_i = (\lambda_{i+1} + \boldsymbol{o}_{i+1}) - (\lambda_i - \boldsymbol{o}_i) \geq \varepsilon$. The total constrained optimization problem can be written as:

$$\begin{aligned} \min_{\boldsymbol{o}} \quad & \frac{1}{2}\|\boldsymbol{o}\|_2^2 \\ \text{subject to} \quad & \boldsymbol{o}_{i+1} - \boldsymbol{o}_i \geq \varepsilon - (\lambda_{i+1} - \lambda_i) \end{aligned} \tag{30}$$

The $(n-1)$ inequality constraints are linear and can be written in matrix-vector form. To further ensure that the total range of the eigenvalues is not changed, the equality constraints $\boldsymbol{o}_0 = \boldsymbol{o}_n = 0$ can be added. As an initial guess, the offsets can be set to equally separate the eigenvalues in their range, which is guaranteed to satisfy all constraints. The optimal solution $\boldsymbol{o}^*$ can be calculated efficiently using constrained optimization.

In conclusion, using the slightly perturbed Laplacian $\hat{\boldsymbol{L}}_{sym} = \boldsymbol{L}_{sym} + \boldsymbol{U}\text{diag}(\boldsymbol{o}^*)\boldsymbol{U}^{\mathsf{T}}$ as input to the eigendecomposition in the forward pass results in usable gradient via back-propagation. Note that to get the perturbation, the eigendecomposition of the original Laplacian has to be computed. Thus, it can be checked for the presence of repeated eigenvalues, and a second perturbed eigendecomposition is only computed when necessary. Tab. 3 includes results using this approach, which seems to work about as well as the perturbation approximation.

