# OpenReview forum: "Adversarial Robustness of Graph Transformers"
_ICLR.cc/2025/Conference — Submitted to ICLR 2025_

### Official Review · Reviewer_BEc6 · 2024-10-25

**Soundness:** 3
**Presentation:** 3
**Contribution:** 1
**Rating:** 5
**Confidence:** 4

**Summary:**

This abstract discusses the vulnerability of Graph Transformers (GTs) to adversarial attacks, contrasting them with Message-Passing Graph Neural Networks (MPNNs), which are known to be susceptible. Despite GTs' growing importance, their robustness is underexplored. The authors design adaptive attacks specifically targeting GTs by creating gradient-based attacks focused on structural perturbations. They evaluate these attacks across five popular GT architectures, examining how GTs handle attention mechanisms and positional encodings (PEs). Results indicate that GTs are extremely fragile in many scenarios. To address this, the study also explores how adaptive attacks can inform adversarial training to enhance robustness.

**Strengths:**

1. This work provides an adaptive gradient-based structure attacks for GTs.
2. This work is the first to study the adversarial robustness of graph transformers
3. The presentation and writing are very clear and convincing

**Weaknesses:**

1. The main weakness of this paper is the limited contribution. The adaptive attack is specifically designed for a particular defense, which limits its generalizability. It would be a significant improvement if the adaptive attacks could share a generalizable principle across all GTs.

2. As shown in the experiments, the relaxation technique is ineffective, and the adaptive attack is not strong enough. In Figures 3, 4, 5, and 6, there are many cases where the adaptive attack is weaker than the transfer attacks. In the ablation study (Tables 1,2 and 3), relaxing the non-differentiable operator does not achieve a better attack in some cases and only provides minimal improvement in others. Additionally, the experiments do not compare results with non-adaptive gradient-based attacks. As noted in the paper, "A good approach might be to
try all individually as an ensemble and choose the strongest. " This work, however, appears highly incremental and lacks universality. All of these findings raise concerns regarding the effectiveness of the proposed method.

3. The attack baselines are limited; the experiments only compare transfer attacks and random attacks, which is insufficient. Furthermore, the adaptive attacks do not seem stronger than the transfer attacks. Additional baselines, including metattack and nettack, should be incorporated.

**Questions:**

See the weaknesses.

---

> ### Author Response · Authors · 2024-11-27
> **Response to BEc6**
>
> We thank the reviewer for their feedback and acknowledgment of the good presentation!
>
> ### W1: Limited contribution because attacks are not general
>
> Adaptive attacks are necessarily model-specific. Nonetheless, our adaptive attacks are applicable to any model that uses one of the components that we relaxed. Furthermore, we provide *general guidelines* for designing such adaptive attacks for graph transformers. Note that our attack perturbations can be universally applied to any model as transfer attacks (in the same way, established attacks such as Nettack and Mettack are universally applicable). Due to the strength of **our** "best transfer", one can use the perturbations of our work for a first test on other GT's robustness. We provide the detailed discussion and reasoning in the general comment, "Universal adaptive attacks do not exist."
>
> Since universal adaptive attacks are not attainable, we believe this should not be considered a weakness of our work. However, if the reviewer is not fully convinced by our arguments, we would kindly ask for more clarification so that we can effectively address the reviewer's concern.
>
>
> ### W2: Attacks too weak
>
> If comparing to the state of the art, **our adaptive attacks are usually the strongest (in 88\% of the cases for budgets of 10% or higher) and often by a large margin (e.g. in Figure 6, our adaptive achieves 1% adversarial accuracy vs. 60% of second-best attack)**. Our results clearly demonstrate that **prior adversarial attacks are grossly insufficient for the evaluation of adversarial robustness of graph transformers**. We refer to our 'Effectiveness of our attacks' global comment for more details, also contrasting the previous with the updated presentation in the paper.
>
> Regarding the ablations, note that for Graphormer and SAN at least one relaxation is necessary to compute the gradients. The fact that the attack sometimes works better when only relaxing a single component is not entirely unexpected. As suggested in the *Recurring Attack Theme T2* of \[2\]:
> > only one or two \[components\] would be sufficient for the defense to fail. Targeting only these components can lead to attacks that are simpler, stronger, and easier to optimize.
>
> \[2\] F. Tramèr, N. Carlini, W. Brendel, "On Adaptive Attacks to Adversarial Example Defenses", NeurIPS 2020.
>
> ### W3: Attack baselines
>
> We thank the reviewer for the suggestion on further attack baselines. In the revision, we have now added **two new baseline attacks** including the "GCN PRBCD transfer" attack as a (state-of-the-art) baseline. Note that the "GCN PRBCD transfer" has been shown to be more effective than SGA and Nettack \[5\]. To the best of our knowledge, there has been no applicable attack that is significantly/consistently stronger in the meantime.
>
> As we explain in our 'Requested baselines' global comment, many suggested baseline attacks (including Mettack and Nettack) are not suitable for the (GT "preferred") datasets and attack settings we evaluate on.
>
> If our rebuttal did not fully address the points brought up by the reviewer, we kindly ask for clarification so we can effectively address the reviewer's concerns.
>
> \[5\] S. Geisler, T. Schmidt, H. Şirin, D. Zügner, A. Bojchevski, S. Günnemann, "Robustness of Graph Neural Networks at Scale", NeurIPS 2021.

---

> > ### Comment · Reviewer_BEc6 · 2024-11-28
> >
> > Thanks for your feedback. I will raise the score accordingly.

---

> > > ### Author Response · Authors · 2024-11-28
> > >
> > > We thank the reviewer for their response and for raising the score! Nevertheless, we would appreciate feedback on what weaknesses remain s.t. we can better convert the efforts on the review into an improved paper!

---

### Official Review · Reviewer_FPkQ · 2024-10-25

**Soundness:** 2
**Presentation:** 2
**Contribution:** 3
**Rating:** 5
**Confidence:** 3

**Summary:**

This paper presents the first analysis of the adversarial robustness of Graph Transformers (GTs). The authors design adaptive gradient-based attacks for five representative GT architectures, including those with specialized attention mechanisms and Positional Encodings (PEs) based on random walks, pair-wise shortest paths, and the Laplacian spectrum. The results reveal that GTs can be catastrophically fragile to adversarial attacks, with even slight perturbations to the graph structure leading to significant performance degradation. Consequently, the authors show how to leverage their adaptive attacks for adversarial training, substantially improving the robustness of GT models.

**Strengths:**

- This is the first study on the adversarial robustness of Graph Transformers (GTs), which have become popular alternatives to Message-Passing Graph Neural Networks (MPNNs).
- The authors design adaptive gradient-based attacks for five representative GT architectures, including those with specialized attention mechanisms and Positional Encodings (PEs).
- The results show that GTs can be vulnerable to adversarial attacks, with even small perturbations to the graph structure leading to significant performance drops.
- The authors use their adaptive attacks to develop an effective adversarial training strategy that substantially improves the robustness of GT models.

**Weaknesses:**

- The mathematical formulas in the paper seem to appear out of nowhere, lacking sufficient explanation and intuition. For instance, in Eq. (6) ($\hat{w} = w + \log{A}$), the paper does not explain why the log function is necessary. Does the log function have special properties here? Providing a theoretical analysis to explain why such a design could help adversarial attacks would be very beneficial. The same issue arises in Eq. (7) (8) (9) (10).
- In Figure (1b), the attack effect for Graph Transformer （Graphormer, GRIT, SAN）seems not to be very well. Under small budget (for example, 5%), the accuracy drops are not very much compared with Figure (1a) and (1d). Could the author provide a further explanation?
- The writing is so confusing and needs to be improved. For example, the author directly use $deg(v_i)$ in Eq. (7) without any definition.
- The experiment setting has some problems. First, this paper lacks adversarial attack baselines. They should include both typical adversarial attacks and adversarial attacks for graphs. There should be many existing works, like [1, 2].
- If the authors want to validate the effectiveness of adversarial training, they should also compare it with typical graph adversarial defense methods, like [3]. In total, the experiments of this paper are obviously insufficient.
- The clarification of the experiment setting is also confusing. The authors list three different attack methods, but I cannot tell which one is proposed by this paper.

[1] Adversarial Attack on Graph Neural Networks as An Influence Maximization Problem

[2] Graph Backdoor

[3] GNNGuard: Defending Graph Neural Networks against Adversarial Attacks

**Questions:**

The same as the weaknesses

---

> ### Author Response · Authors · 2024-11-27
> **Response to FPkQ**
>
> We thank the reviewer for the review and, particularly, feedback on improving out work!
>
> ### W1: Formulas not explained
>
> We have updated and moved the corresponding section to more clearly explain the reasoning behind the log-term bias in the local attention relaxation. The revised section (shown in blue) now includes equations 6 and 7.
>
> We hope that the improved explanation for this case helps in understanding the other equations as well. More details on the GT architectures, as provided in the appendix C, would most likely also be helpful to fully understand the relaxations.
>
> ### W2: Robustness difference between datasets
>
> It is very challenging to say a priori why and in which setting a model is robust. This is one of the reasons why adaptive attacks like ours are vital for anyone who seriously wants to deploy a GT and, thus, needs to understand the limitations in their respective setting.
>
>
> ### W3: Confusing writing
>
> We want to point out that we do define the node degree $\mathcal{deg}(v_i)$ in the first paragraph of section 2 (originally line 78). If there are other parts of the paper where the confusing writing could be improved, we would appreciate suggestions.
>
> ### W4: Attack baselines
>
> We thank the reviewer for pointing out the matter of attack baselines. In the revision, we have now added two new baseline attacks including the state-of-the-art "GCN PRBCD transfer" attack. To the best of our knowledge, there has been no applicable attack that is significantly/consistently stronger in the meantime.
>
> While there are many other existing works, not all apply to the same settings and are thus not comparable to each other. For instance:
>  - The first suggested attack (InfMax) only perturbs the node features while the graph structure remains the same. We are interested in the opposite case, where only the graph structure is modified, which affects the positional encodings used in graph transformer models.
>  - The second suggested attack (graph backdoor) corresponds to a different threat model, where the attack already occurs at training time. In the *evasion* setting that we consider, the models have already been trained on clean data (fixed weights), and the test input is perturbed.
>
> Thus, these two attacks would not be meaningful baseline comparisons. For further details, we kindly refer to our **'Requested baselines'** global comment. However, we are happy about specific suggestions that are applicable to our setting (and ideally go beyond transferring perturbations from a GCN).
>
> ### W5: Defense baselines
>
> The main contributions of our paper are the adaptive attacks for graph transformers and robustness evaluations. However, we also demonstrate that our attacks can be used for Adversarial Training (AT) to improve the robustness. We believe these results are not trivial and may be of interest to further research. Due to the many aspects like inductive vs. transductive, node-level vs. graph-level, etc. a thorough/meaningful comparison on when GNNs (+ defenses) are superior to GTs requires multiple works on its own.
>
> As a consequence of not comparing to baselines, we do not claim SOTA results or improvements over alternative defenses. However, it has been shown by \[4\] that most existing GNN defenses can be effectively be broken by adaptive attacks such as ours. In contrast, our AT Graphormer models are impressively robust against our adaptive attacks.
>
> \[4\] F. Mujkanovic, S. Geisler, S. Günnemann, A. Bojchevski,  "Are Defenses for Graph Neural Networks Robust?", NeurIPS 2022.
>
> ### W6: Experiment setting
>
> In the revision, we have updated the figures and captions to explain the attack settings more explicitly and highlight better our main attack. For more details on the baselines and which results are from our attacks, we kindly refer to our 'Effectiveness of our attacks' global comment.

---

> > ### Comment · Reviewer_FPkQ · 2024-12-02
> > **Response to authors**
> >
> > Thanks for your detailed response. Despite the good explanations given by the authors, I am not fully convinced. I chose to keep my original score.

---

> > > ### Author Response · Authors · 2024-12-02
> > >
> > > We thank you for the response! Would you mind briefly detailing the remaining reservations? We would very much like to be able to translate the review into further improvements.

---

### Official Review · Reviewer_yZ1C · 2024-11-01

**Soundness:** 3
**Presentation:** 2
**Contribution:** 2
**Rating:** 6
**Confidence:** 3

**Summary:**

This work explores the adversarial robustness of five graph transformer (GT) models. The author designs adaptive gradient-based attacks on five GT models to poison the model’s node and graph classification performance. This work is the first to show that GTs are commonly vulnerable to adversarial attacks and proposes leveraging adversarial training to improve GTs’ adversarial robustness.

**Strengths:**

1. This paper establishes a good baseline for gradient-based adversarial attacks on graph transformers.

2. The work solves the problem of gradient-based structure attacks on GTs by introducing relaxation to attention scores.

3. The author has done explicit experiments on five graph transformers and types of attacks to validate the attack's effectiveness.

**Weaknesses:**

1. There are some typos and explanations which are not clear. For example, the last adjacency matrix symbol in Equation (6) should be \tilde{A} not A. The author may explain what ‘H’ means in equation (3) as it was not shown or explained before. Also, the author may clarify that the adjacency matrix ‘A’ in equation (2) is already self-looped.

2. The author explains different types of GTs in detail in the background section. However, the author may also need to clarify whether the attack is aimed at degrading performance on training data, test data, or both, and to provide a formal definition of the attack loss function used in their experiments.

**Questions:**

1. Can the author provide concrete examples of potential real-world scenarios where these attacks might be relevant or impactful for each type of graph transformer discussed in the paper?

2. The author mentioned that the model output is discontinuous because positional encoding and attention mechanisms in GTs are designed for a discrete graph structure at the beginning of Section 3. Does this mean PEs and ATs are discontinuous because the adjacency matrix is discrete? Normally, the adjacency matrix becomes continuous values when you do the attack. Does this naturally solve the problem that PEs and ATs are discontinuous, so you do not need the relaxation on attention scores?

3. The author claimed in Principle II that the relaxed model function f shoule be continuous w.r.t continuous version of A. Can the gradient-based attack be conducted if it is continuous but not differentiated?

4. Can the author explain why introducing relaxation by adding the log of continuous edge probability to the attention score in equation (6) can solve the gradient problem? What’s the reason for this specific relaxation?

5. Is the method only adapted to PGD and PRBCD attacks? Has the author considered adapting it to other attacks like meta-attacks [1]? Can the author show some results?

[1] https://openreview.net/pdf?id=Bylnx209YX

---

> ### Author Response · Authors · 2024-11-27
> **Response to yZ1C**
>
> We thank the reviewer for their feedback and for acknowledging that our work is a good basis for adversarial attacks on GTs.
>
> ### W1: Notation and clarity
>
> We thank the reviewer for pointing out these improvements, we have addressed them in the revision (modifications are shown in blue).
> For GTs, the adjacency matrix is not self-looped (and adding a self-connection is not considered a valid attack perturbation, such that the diagonal always stays zero).
>
> ### W2: Clarity on attack setting
>
> We thank the reviewer for the feedback and have added a reminder of the attack setting to the evaluation section and all figures to improve clarity. However, we would like to point out that we did specify the attack setting in sub-section 2.1 (originally line 86):
> > untargeted white-box evasion attacks, i.e., an attacker with full knowledge of model and data attempts to change the trained model’s prediction to any incorrect class at test time by slightly perturbing the input graph structure. For node-level tasks we focus on global attacks that minimize the overall performance metric across all nodes.
>
> We previously described the attack losses in section 5 (originally line 317). In the revision, we have modified and moved this to section 2.1 for increased clarity.

---

> ### Author Response · Authors · 2024-11-27
> **Response to questions**
>
> ### Q1: Real-world scenarios
>
> One scenario is that of fake news detection \[A\] as we explore with the fake news detection datasets (UPFD). A model could be deployed to automatically detect if a social media post contains fake news. An adversary, similar to our attack, then might access different accounts under their control to retweet the post with the goal of deceiving the fake news detection. This is similarly explored for other GNN-based fake news detectors in \[B\]. Another important application of graph learning methods is spam detection \[C\], where structure perturbations can model spammers adapting their posting strategy to avoid detection \[D\].
>
> Nevertheless, due to the white-box attack setting, we are mostly concerned with understanding the limitations of a model and less with IT security. Understanding the limitations of a model is critical for all wide-spread applications. To what extent GTs are the better choice for applications is not up to us. However, as GTs are showing strong performance in benchmarks, it is not unreasonable to assume that they could be deployed. The question we tackle relates to the simple and important question: are the predictions of graph transformers more or less fragile than those of MPNNs? To answer such a question, the field requires the tools we study.
>
> \[A\] Hu et al. "An overview of fake news detection: From a new perspective", Fundamental Research 2024
>
> \[B\] Wang et al. "Attacking fake news detectors via manipulating news social engagement", WWW 2023
>
> \[C\] Li et al. "Spam Review Detection with Graph Convolutional Networks", CIKM 2019
>
> \[D\] Soliman et al. "AdaGraph: Adaptive Graph-Based Algorithms for Spam Detection in Social Networks", 2017
>
> ### Q2: Discontinuity
>
> For some models, such as GRIT, a continuous change of the values of the adjacency matrix does indeed result in a continuous change of the output. However, other models are "designed for a discrete graph structure" to such an extent that the architecture and implementation rely on the certainty that the values in the adjacency matrix are discrete (e.g., for Graphormer). When inputting continuous values in the adjacency matrix, the computation becomes undefined and trying to execute the original implementation of the model results in an error.
>
> ### Q3: Relaxation differentiability
>
> For the relaxations we only require continuity and not differentiability everywhere. While it would be nice if $\tilde{f}\_{\theta}$ was continuously differentiable, it would be challenging to design it in such a way. Nonetheless, for optimization, it should suffice for the function to be continuous. As we stated directly after the principles:
> > While Principle II might appear surprising at first glance, we argue that we do not need to enforce stronger standards on $\tilde{f}\_{\theta}$ than perhaps the most widely used activation function ReLU
>
> (Since NNs using ReLU can evidently be optimized as well).
>
> ### Q4: Local attention relaxation
>
> We have updated and moved the corresponding section to more clearly explain the reasoning behind the log-term bias in the local attention relaxation. The revised section (shown in blue) now includes equations 6 and 7.
>
> ### Q5: Other attacks
>
> Since our relaxations enable the computation of gradients w.r.t. the adjacency matrix in general, we are not limited to using PDG/PRBCD. Other options for gradient-based attacks are e.g. FGSM/FGA, and GRBCD \[4,5\]. Moreover, \[5\] (and \[4\]) show that for adaptive global evasion attacks PRBCD (and PGD) tend to perform the best. Therefore, we would not expect the results and insights to change much by adding a new gradient-based attack such as GRBCD. For this reason, we did not pursue this further.
>
> We also want to note that we apply evasion attacks (test-time); using a poisoning attack (training time) such as Mettack is not directly applicable.
>
> \[4\] F. Mujkanovic, S. Geisler, S. Günnemann, A. Bojchevski,  "Are Defenses for Graph Neural Networks Robust?", NeurIPS 2022.
>
> \[5\] S. Geisler, T. Schmidt, H. Şirin, D. Zügner, A. Bojchevski, S. Günnemann, "Robustness of Graph Neural Networks at Scale", NeurIPS 2021.

---

### Official Review · Reviewer_t2zt · 2024-11-03

**Soundness:** 3
**Presentation:** 4
**Contribution:** 3
**Rating:** 6
**Confidence:** 4

**Summary:**

This paper investigates the robustness of Graph Transformers (GTs) against adversarial attacks, a relatively unexplored area compared to the vulnerabilities of Message-Passing Graph Neural Networks (MPNNs). The authors propose adaptive, gradient-based structure attacks targeting GT architectures and evaluate these attacks across different attention mechanisms and positional encoding methods. Their findings reveal that GTs can be highly vulnerable to adversarial attacks, similar to MPNNs. They further demonstrate that adversarial training effectively enhances the robustness of GTs, providing a defense mechanism against these vulnerabilities.

**Strengths:**

1.	The paper is the first to propose adaptive gradient-based attack strategies specifically for Graph Transformers (GTs), establishing a foundation for studying the adversarial robustness of GTs.
2.	The authors conduct multiple experiments across multiple tasks and threat models, demonstrating GTs’ performance under different attack budgets and highlighting their vulnerabilities to adversarial attacks.
3.	The authors propose an adversarial training strategy that reduces the hypersensitivity of some GT architectures, effectively enhancing their adversarial robustness and showing the potential of adversarial training in GTs.
4.	The paper is organized well, and the logic is smooth and convincing.

**Weaknesses:**

1.	The attack approach lacks generalizability and is overly empirical, requiring specific design adjustments for each model rather than presenting a systematic method or architecture, making it more of an empirical study than a universal solution.
2.	The effectiveness of the "adaptive attack" is questionable, as it does not significantly outperform grey-box attacks. Non-adaptive attacks without relaxation should have been included for comparison, and the baseline comparisons are limited to weak black-box attacks (random and transfer), which diminishes the impact of the results.
3.	The selection of baselines for both attacks and defenses is insufficient. The study lacks comparisons with established attack methods like Metattack and Nettack, as well as with more diverse defense strategies, such as other robust training techniques, graph cleaning methods, detection techniques, etc.

**Questions:**

1.	Could the authors include more attack and defense baselines?
2.	What is the performance of gradient-based attacks without relaxation?

---

> ### Author Response · Authors · 2024-11-27
> **Response to t2zt**
>
> We thank the reviewer for their critical thoughts on improving our paper!
>
> ## W1: Universal adaptive attacks
>
> Adaptive attacks are necessarily model-specific, and their development contains empirical parts. Nonetheless, our adaptive attacks are applicable to any model that uses one of the components that we relaxed. Furthermore, we provide *general guidelines* for designing such adaptive attacks for graph transformers. Note that our attack perturbations can be universally applied to any model as transfer attacks (in the same way, established attacks such as Nettack and Mettack are universally applicable). Due to the strength of **our** "best transfer," one can use the perturbations of our work for a first test on other GT's robustness. We provide the detailed discussion and reasoning in the general comment, "Universal adaptive attacks do not exist."
>
> Since universal adaptive attacks are not attainable, we believe this should not be considered a weakness of our work. However, if the reviewer is not fully convinced by our arguments, we would kindly ask for more clarification so that we can effectively address the reviewer's concern.
>
> ## W2 (and Q1): Attack effectiveness and baselines
>
> If comparing to the state of the art, **our adaptive attacks are usually the strongest (in 88\% of the cases for budgets of 10% or higher) and often by a large margin (e.g. in Figure 6, our adaptive achieves 1% adversarial accuracy vs. 60% of second-best attack)**. Furthermore, we want to point out that we do include *"non-adaptive attacks without relaxation"*: the (brute force) random attack and the GCN transfer attack. These attacks are the applicable state of the art in the GNN attack domain. We refer to the general comment for an in-depth discussion, and we have changed our presentation to convey this better.
>
> ## W3 (and Q1): Not enough attack/ defense comparisons
>
> We appreciate the suggestions on incorporating further baselines. We now include **two additional** baseline attacks. However, it is not trivial to find other suitable baseline attacks for the datasets and attack settings we study (e.g., this is why Mettack and Nettack are not suitable). We are happy about specific suggestions that are applicable to our setting (and ideally go beyond transferring perturbations from a GCN). For more details on this point, please see 'Requested baselines' in our general comment.
>
> The main contributions of our paper are the adaptive attacks for graph transformers and robustness evaluations. However, we also demonstrate that our attacks can be used for Adversarial Training (AT) to improve their robustness. We believe these results are not trivial and may be of interest to further research. Due to the many aspects like inductive vs. transductive, node-level vs. graph-level, etc., a thorough/meaningful comparison of when GNNs (+ defenses) are superior to GTs requires multiple works on its own.
>
> As a consequence of not comparing to baselines, we do not claim SOTA results or improvements over alternative defenses. However, it has been shown by \[4\] that most existing GNN defenses, such as GNNGuard \[9\], can be effectively broken by adaptive attacks such as ours. In contrast, our AT Graphormer models are impressively robust against our adaptive attacks.
>
> ## Q2: Gradient-based without relaxations
>
> The results for all transfer attacks can be considered as results for gradient-based attacks that do not use the relaxations of the victim model.
> Regarding adaptive attacks, for models that only include the adjacency matrix in a non-differentiable manner (e.g., Graphormer), at least one relaxation is required to compute the gradient. Without any relaxation, the gradient is undefined and, e.g., PyTorch will throw an error. Thus, no adaptive gradient-based attack is possible. However, in the ablations, we show how using different relaxations individually impacts the performance.
>
> We are confident that this addresses all the concerns of the reviewer and are looking forward to their feedback!

---

> > ### Comment · Reviewer_t2zt · 2024-12-02
> > **Increase my score**
> >
> > Dear authors,
> >
> > Thanks for the detailed response. I will increase my score.

---

### Author Response · Authors · 2024-11-27
**Global Response**

We thank the reviewers for their helpful feedback to improve our work! Based on it, we now include interesting **new results and discussions**. However, we believe that the scores/feedback of some reviewers are guided strongly by some misunderstandings:
1) It is an impossible desiderata to request universal adaptive adversarial attack (that are efficient), as we argue in detail in the **'Universal adaptive attacks do not exist'** comment below. Thus, that our attacks do not capture all possible models/graph transformers should not be regarded as a weakness of our work, it is rather a general characteristic of adversarial attacks.
1) As we detailed previously in ll. 311-314, our transfer attacks are much stronger than the usual definition of transfer attacks, leading to doubts regarding the effectiveness of our adaptive attacks. In the revision, we moved the transferability analysis to a separate paragraph and figure (Fig. 7) with an improved explanation of the (relabeled) "best transfer" results. Furthermore, we added two new baselines to the main result figures (Fig. 3-6) that provide better context.  In the **'Effectiveness of our attacks'** comment, we explain in detail why our transfer attacks are already much stronger than commonly reported transfer attacks and that in comparison to a single transfer attack, our adaptive attacks are almost consistently stronger and often by a large margin.
5) Most attack baselines are incompatible with our setting, that is tailored to graph transformers. Nevertheless, we added two new baselines ("GCN PRBCD transfer" and "Random perturbation"). In the **'Requested baselines'** comment below, we explain in detail why certain baselines are not applicable in our evaluation setting.

To support these main points, we now provide a **revision** of our paper (uploaded). Notable modifications are shown in blue and include:
- **Two new attack baselines** (updated Fig. 3-6).
- Additional specification of the attack settings and threat models.
- Improved explanations for equations and notation.
- A separate analysis of the (best) transfer attacks (Fig. 7, and 11-14).

Given these major clarifications, we would **appreciate a reevaluation of our work**.

---

> ### Author Response · Authors · 2024-11-27
> **Universal adaptive attacks do not exist**
>
> **Why use adaptive attacks?**: While the purpose of using non-differentiable components in GTs is not to defend against adversarial attacks, the resulting effect is the same as for defenses that rely on gradient obfuscation \[1\]. Simply applying attacks that were designed for a different setting is likely to result in estimates that are too optimistic. For example, the evaluation with adaptive attacks sometimes even revealed that defenses might harm the robustness while the defenses helped with non-adaptive attacks \[4\]. Therefore, adaptive attacks are needed to study the robustness of these GTs more accurately. Our continuous relaxations are analogous to the Backward Pass Differentiable Approximations proposed in \[1\], although we also use the relaxations in the forward pass.
>
> **Adaptive attacks are always specific and empirical**: As stated in the abstract of \[2\]:
> > no single strategy would have been sufficient for all defenses [...] adaptive attacks cannot be automated and always require careful and appropriate tuning to a given defense
>
> Thus, studies employing adaptive attacks are necessarily model-specific and empirical by trying different attack strategies tailored to specific models or defenses in order to find the strongest adversarial examples \[1,2,3,4\] (if an exhaustive search is infeasible as it is the case already for small graphs). However, we do provide **general design principles** to develop adaptive attacks for graph transformers (and non-differentiable components in general). Furthermore, our specific relaxations can provide useful examples for similar components.
>
> **Applicability to other GNNs**: Many of our findings can be used to attack other GNNs and GTs. For example, PEs are popular choices for different GNNs. As suggested in the *Recurring Attack Theme T2* of \[2\]:
> > only one or two \[components\] would be sufficient for the defense to fail. Targeting only these components can lead to attacks that are simpler, stronger, and easier to optimize.
>
> Conversely, from a robustness perspective, every component requires its own study regarding its robustness properties as it might introduce another vulnerability. This is further supported by our ablation study, indicating that any GNN using just one of these components can be attacked using our proposed relaxations.
>
> **Established attacks are general because they are transfer attacks**: In fact, most established GNN structure attacks, such as Nettack \[6\] and Mettack \[7\], are general and universally applicable for this reason: They are non-adaptive transfer attacks computed for a surrogate model. This is also described in \[4\]:
> > Such transfer attacks are so common in the graph domain that their usage is often not even explicitly stated, and we find that the perturbations are most commonly transferred from Nettack or Metattack (both use a linearized GCN). Other times, the authors of a defense only state that they use PGD (aka “topology attack”) without further explanations. In this case, the authors most certainly refer to a PGD transfer attack on a GCN proxy.
>
> In summary, adaptive attacks always require model-specific modifications/considerations. Our adaptive evaluation of graph transformers is already much more general than impactful works like Nettack \[6\] and Mettack \[7\] as they solely study (linearized) GCNs. However, prior work usually appears generally applicable since they only transfer perturbations. Thus, it should not be considered a weakness that adaptive attacks are model-specific. That adaptive attacks are model-specific is a general property of adversarial attacks and cannot be attributed to our work specifically. Nevertheless, for widespread applicability, we study a representative sample of GTs, spanning many frequently used components.

---

> ### Author Response · Authors · 2024-11-27
> **Effectiveness of our attacks**
>
> We use adaptive attacks to attack the victim models directly because this often results in stronger attacks and, therefore, better model robustness estimates. Indeed, if compared to the state of the art, **our adaptive attacks are usually the strongest (in 88\% of the cases for budgets of 10% or higher) and often by a large margin (e.g. in Figure 6, our adaptive achieves 1% adversarial accuracy vs. 60% of second-best attack)**.
>
> However, we can (and do) also use the perturbations generated by our adaptive attacks as transfer attacks for other models (previously "transfer" and now "best transfer"). These transfer attacks are, of course, completely general and can be applied to any model of arbitrary type. Notably, the strength of **our** "best transfer" attacks seems to have led to some **misinterpretation**, suggesting that our adaptive attacks are weak in comparison. However, since the transfer attacks are generated by our own adaptive attacks for different graph transformers, the strength of these transfer attacks is, of course, intimately coupled to the strength of our adaptive attacks. Moreover, we transfer the results of the (up to) 8 adaptive attacks for different models, where we take the strongest perturbation for each instance. In other words, the *"best transfer"* attack can be considered an ensemble attack. We hypothesize that the variety of the transfer attacks substantially benefits the effectiveness. In contrast, we solely report the results for a single adaptive attack. We originally included "best transfer" mainly to show the transferrability of the attacks between GT models and to get the worst-case perturbations for the best robustness estimates.
>
> To disentangle the presentation and first compare our adaptive attacks to the prior state of the art, we now report in Figure 3-6 our adaptive attacks in comparison to a "GCN PRBCD transfer" attack (new baseline/state of the art/common practice). This clearly demonstrates that **prior adversarial attacks are grossly insufficient for evaluating the adversarial robustness of graph transformers**. Also, note that the "GCN PRBCD transfer" is not a weak baseline as it has been shown to be more effective than GCA and Nettack \[5\]. To the best of our knowledge, there has been no applicable attack that is significantly/consistently stronger in the meantime.
>
> Leveraging the variety of our adaptive attacks on different representative graph transformers, we can use them as a diverse transfer attack to new architectures. In Figure 7, we now show that our diverse "best transfer" already constitutes a very effective attack in many but not all settings. Our "best transfer" could be used for future graph transformers as a "unit test." In other words, if a graph transformer is already non-robust w.r.t. "best transfer," then one might skip the considerable effort for designing adaptive attacks.
>
> Based on the above reasoning, our transfer attacks could be considered more general than those of Nettack and Mettack (considering our relaxed GT models as surrogate models) since we allow the transfer of the results from multiple victim models. Admittedly, this is not the most computationally efficient approach, as attacking GTs is more expensive than attacking a GCN.
>
> In summary, prior work is severely insufficient for the robustness evaluation of GTs. Moreover, the strength of "best transfer" is mostly a consequence of the strength of our adaptive GT attacks and should not be considered a weakness of our paper!

---

> ### Author Response · Authors · 2024-11-27
> **Requested baselines**
>
> We now additionally include "GCN PRBCD transfer" and "random perturbation" as baseline attacks. The GCN PRBCD transfer attack follows the same principle as many other popular (and requested) baselines: a gradient-based attack (PRBCD) on a "simpler" surrogate (GCN) that gets transferred to the victim model. Moreover, it has been show to be just as effective (or more) than other baselines \[5\].
>
> We want to note that it is not easy to find suitable baseline attacks for our learning task and attack settings. Admittedly, we only mentioned these settings in the background section, which presumably led to some misunderstanding. In the revision, we have made the setting much clearer in all relevant sections and figures (shown in blue). Below, we explain in detail why it is not possible to employ most of the baselines requested by the reviewers.
>
> Specifically, we evaluate on datasets that contain many graphs (of smaller size) where the task is *inductive* graph (or node) classification because, in such settings, GTs perform best. Existing GNN robustness literature often studies the *transductive* node classification task on a single (larger) graph. However, most GTs are not applied to these tasks as they do not perform well in this setting. For the inductive setting, *evasion* (test-time) attacks are of most interest; while for the transductive learning setting, *poisoning* attacks are often more appropriate \[4\]. Thus, for our setting we focus on evasion attacks.
>
> **Incompatible attack baselines**: Due to our specific tasks and settings, the following types of attacks are not suited as baselines for our evaluations:
>
> - **Feature attacks**: Our attacks only perturb the graph structure, not the node features. Adding feature attacks as baselines would not be very meaningful, as they are not directly comparable (e.g. InfMax \[10\]).
> - **Different threat models**: Since we only evaluate evasion attacks (on inductive datasets), requested baselines that correspond to fundamentally different threat models, such as training-time *poisoning* (e.g. Mettack \[7\]) or *backdoor* (e.g. Graph Backdoor \[11\]) attacks are not meaningfully comparable.
> - **Local attacks (for node classification)**: Some popular attacks, such as Nettack \[6\] and SGA \[8\], are specifically for *local* attacks, i.e., the prediction of a single node in a node classification task is attacked. The only node classification dataset we evaluate on is CLUSTER. However, we evaluate global attacks where the overall classification accuracy over all nodes is degraded. While we could add a local attack evaluation to compare to the above-mentioned baselines, the global attack is already extremely effective in assessing this "fragile" data. For CLUSTER, local attacks are even more straightforward (disconnect the target node from its cluster and attach to a different one). Thus, hardly any difference between (all successful) attack methods can be perceived. We welcome suggestions for other (inductive) node classification datasets on which we could evaluate local attacks.
>
> Additionally, our *"random attack"* is a stronger random attack than what is usually reported. Specifically, "random attack" is a brute-force random search attack with a computational budget that matches the adaptive attacks. This results in a much more effective attack than a single random perturbation (which is also often used as a baseline). Because of this, the name may be slightly misleading. To clarify this, we have added the simple single "random perturbation" as an additional baseline to make the difference explicit.
>
>
> # References
>
>
> \[1\] A. Athalye, N. Carlini, D. Wagner, "Obfuscated Gradients Give a False Sense of Security: Circumventing Defenses to Adversarial Examples", ICML 2018.
>
> \[2\] F. Tramèr, N. Carlini, W. Brendel, "On Adaptive Attacks to Adversarial Example Defenses", NeurIPS 2020.
>
> \[3\] N. Carlini, D. Wagner, "Towards Evaluating the Robustness of Neural Networks", SSP 2017.
>
> \[4\] F. Mujkanovic, S. Geisler, S. Günnemann, A. Bojchevski,  "Are Defenses for Graph Neural Networks Robust?", NeurIPS 2022.
>
> \[5\] S. Geisler, T. Schmidt, H. Şirin, D. Zügner, A. Bojchevski, S. Günnemann, "Robustness of Graph Neural Networks at Scale", NeurIPS 2021.
>
> \[6\] D. Zügner, A. Akbarnejad, S. Günnemann, "Adversarial Attacks on Neural Networks for Graph Data", KDD 2018.
>
> \[7\] D. Zügner, S. Günnemann, "Adversarial Attacks on
> Graph Neural Networks via Meta Learning", ICLR 2019.
>
> \[8\] J. Li, T. Xie, L. Chen, F. Xie, X. He, Z. Zheng, "Adversarial Attack on Large Scale Graph", TKDE 2021.
>
> \[9\] X. Zhang, M. Zitnik, "GNNGuard: Defending Graph Neural Networks against Adversarial Attacks", NeurIPS 2020.
>
> \[10\] "Adversarial Attack on Graph Neural Networks
> as An Influence Maximization Problem", WSDM 2022.
>
> \[11\] Z. Xi, R. Pang, S. Ji, T. Wang, "Graph Backdoor", USENIX Security 2021.

---

### Author Response · Authors · 2024-12-02
**Discussion Period Closing Soon**

We want to thank all reviewers again for their work and feedback that allowed us to improve the quality of our work. As the deadline to answer to our rebuttal very soon comes to an end, we would greatly appreciate a response if we could resolve the questions and concerns. Otherwise, we are happy to provide further clarifications.

Best regards,
The Authors

---

### Author Response · Authors · 2024-12-03
**Summary of rebuttal phase**

Dear AC, dear reviewers,

We thank again for the feedback on our work and believe that the revision of our manuscript improved as well as addressed the weaknesses. Since graph transformers are gaining importance, we believe that their robustness should not remain unexplored, and we show that graph transformers indeed can be critically vulnerable (sometimes more than message-passing models). However, with the attacks prior to our work, such findings were hardly possible. The changes in our manuscript triggered by the feedback further help us to convey this message!

While we are still awaiting the response of one of the reviewers, none of the other reviewers articulated any remaining weaknesses after our rebuttal, although one reviewer seems not fully convinced for unspecified reasons. As we elaborate in our rebuttal, the main points are:
1. We clarified that adaptive attacks are necessarily model-specific. Both reviewers who criticized this point thereafter increased their scores.
1. We altered the presentation to show the inadequacy of prior work for assessing the robustness of graph transformers and the efficacy of our attacks. Our adaptive attacks are almost consistently more effective (e.g., in 88% of the cases for budgets of 10% or higher) and often by a large margin (e.g. in Figure 6, our adaptive attack achieves 1% adversarial accuracy vs. 60% of second-best attack).
1. Most studies on adversarial robustness/attacks in the graph domain focus on settings where graph transformers do not excel. While the inapplicability of most prior methods prevented us from comparing to some of the requested baselines, it further shines light on how unexplored the studied setting is.

We believe that the matter of the adversarial robustness of graph transformers is a serious one and that our work will be a seminal work for further studies to come. We would appreciate it if the reviewers and area chair would carefully discuss our work once more and, importantly, provide guidance on how to improve if deemed insufficient – particularly important to the field due to the previously unknown fragility of graph transformers in some settings and for the study of strategies to remedy this fragility (like our adversarial training).

---

### Public Comment · ~Lukas_Gosch1 · 2026-03-04
**Revised & Published Paper Version**

The authors want to refer interested readers to [a revised paper version](https://openreview.net/pdf?id=4xK0vjxTWL) accepted at TMLR, awarded with a J2C-Certification, and selected for presentation at ICLR 2026.

Revised Paper Link: https://openreview.net/pdf?id=4xK0vjxTWL

---

### Meta-Review · Area_Chair_Eo8z · 2024-12-20

**Metareview:**

The paper explores the robustness of Graph transformers under adversarial attacks, proposes adaptive, gradient-based structure attacks for GT architectures, and evaluates these attacks against different Graph Transformers. Their findings reveal that Graph Transformers are also vulnerable to adversarial attacks, and leveraging adversarial training can improve GT's adversarial robustness.

Strength:
1. Explore the adversarial robustness of Graph Transformers.

2. Explore the adaptive attacks on different attention methods and position encoding methods in Graph Transformers.

Weaknesses:

1. Lack of baseline for attacks and defenses. Although the authors argue that many other attacks are not in the same setting (like perturbing features), I think it doesn't make sense as the attacker's goal is to fool the GT no matter which method is used. Unless the authors can convince the readers that the other attacks or defenses are not practical.

2. The authors further comment “Most studies on adversarial robustness/attacks in the graph domain focus on settings where graph transformers do not excel”. That also made me concerned about this work's importance, as GT's impact is also not strong enough as it has many problems as the authors mentioned. Furthermore, GT's efficiency is not good on large graphs. Therefore, I think it may not be a good time to evaluate robustness. Maybe the authors can manage to do attacks on Graph LLMs in their revision, its influence will be larger.


Although it's the first work evaluating the adversarial robustness of Graph Transformers, I don't think its findings on GT being vulnerable to attacks is a surprising result. Transformers are not robust on the image, sequential signal, and many other modalities. Therefore, I don't think anyone will believe GT will be robust even if no paper does such experiments. Then I would like to treat this paper as a paper for better attacks. Due to this reason, I think this paper needs comprehensive evaluations.

In summary, I suggest to reject this paper.

**Additional Comments On Reviewer Discussion:**

The authors and reviewers debate about the baseline settings of this paper and I support the reviewers because I think it is just a paper for a better attack.

---

### Decision · Program_Chairs · 2025-01-22

Reject